

# Water flow timing, quantity, and sources in a fractured high mountain

# permafrost rock wall

Matan Ben-Asher[1], Antoine Chabas[2], Jean-Yves Josnin[2], Josué Bock[2], Emmanuel Malet[2], Amaël
Poulain[3], Yves Perrette[2], Florence Magnin[2]

[1] Department of Natural Sciences, Open University of Israel, Ra'anana, Israel

[2] EDYTEM, USMB, CNRS, 5 bd de la mer Caspienne, 73376 Le Bourget du Lac cedex, France

[3] TRAQUA S.A., Namur, Belgium

*Correspondence to*: Matan Ben-Asher (matanbe@openu.ac.il) and Florence Magnin (Florence.Magnin@univ-smb.fr)

## Abstract

Water flow in high mountain rock walls is crucial in landscape evolution and slope stability. However, the timing, quantity, and sources of this flow remain poorly understood. In the Mont Blanc massif, tunnels at the Aiguille du Midi peak (3842 m) provide direct access to steep permafrost-affected rock walls. Over two years (May 2022–October 2023), we monitored water flowing from fractures using a real-time system measuring flow rate, temperature, electrical conductivity, and fluorescent tracers, together with meteorological data and ground surface temperatures.

Results indicate high surface–subsurface connectivity. The water source is primarily snowmelt, with additional inputs from late-summer rainfall. Electrical conductivity, stable isotopes, and recession curve analysis suggest another source of older subsurface ice. Flow onset was closely tied to ATs, with steady diurnal fluctuations appearing once ground surface temperatures exceeded 0 °C. Lag times between daily peaks of flow rate and peaks of air and ground surface temperatures of 3–9 hours and 0–3 hours, respectively, point to rapid unsaturated infiltration conditions. Distinct flow regimes observed in two adjacent fracture systems reflect a complex, heterogeneous network, including sediment-filled fractures with delayed response. Significant flow rate (often >10 L/h) and water temperature often exceeding 5 °C, suggest a significant heat transfer by advection, capable of enhancing permafrost degradation.

This study provides rare direct observations of fracture flow dynamics in steep permafrost rocks, improving understanding of water routing and its response to atmospheric forcing. The findings offer valuable constraints for coupled hydrothermal models, permafrost-related hazard assessments, and the potential impact of climate change.

**Key words**: Permafrost, monitoring, infiltration, Mont Blanc massif, climate change



# 1. Introduction

## 1.1. Hydrogeology of high mountains

Water plays a crucial role in weathering and erosion processes in mountainous landscapes. In the periglacial belt, the presence of water in the shallow subsurface can lead to rock fracturing through ice segregation or volumetric expansion, depending on temperature conditions and saturation levels (Draebing and Krautblatter, 2019; Matsuoka and Murton, 2008). Hydrostatic pressure in undrained fractures is capable of driving catastrophic failure (Hasler et al., 2012; Krautblatter et al., 2013a; Scandroglio et al., 2021). Over geological time scales water is a key catalyst of mechanical rock weathering processes related to subcritical cracking (Eppes and Keanini, 2017). The melting of ice in joints under thawing conditions releases detached blocks and lead to debris and boulders falls, and the formation of scree slopes (Hales and Roering, 2007). Water infiltration in bedrock may also trigger large destabilizations such as rockfalls and rock avalanches by reducing the friction of rough fracture contact surfaces (Krautblatter et al., 2013a). In permafrost ground, the presence of ice sealing in the ground interstices favors the development of high hydrostatic pressures (Fischer et al., 2010; Marcer et al., 2020), possibly enhancing the frequency or magnitude of mass movements. In addition to mechanical pressure, water circulation can also cause thermal perturbations with potential cooling effects in some cases (Maréchal et al., 1999) or warming effects in others (Hasler et al., 2011). In permafrost ground, heat advection from water infiltration could accelerate permafrost degradation (Gruber and Haeberli, 2007; Hasler et al., 2011; Magnin and Josnin, 2021), and potentially develop thawing corridors (Krautblatter and Hauck, 2007; Keushing et al., 2017). Recent observations of increased rock fall activity in high mountains regions was linked with permafrost degradation (Allen et al., 2009; Fey et al., 2025; Gruber et al., 2004; Huggel et al., 2012; Legay et al., 2021; Ravanel et al., 2017; Ravanel and Deline, 2011). The warming of intact frozen rock is commonly related to rockwall destabilization through the decrease of the rock uniaxial and tensile strength (Dwivedi et al., 1998; Krautblatter et al., 2013a; Li et al., 2003; Mellor, 1973; Scandroglio et al., 2025). Water related processes were suggested as a potential mechanistic driver of several rock fall events (Cathala et al., 2024; Erismann and Abele, 2001; Scandroglio et al., 2021; Strauhal et al., 2016). However, while hydrogeological studies in alpine permafrost have primarily focused on coarse-grained terrains such as rock glaciers and scree slopes, little is known about water dynamics within bedrock rockwalls—despite their critical role in slope stability and landscape evolution.

## 1.2. Existing knowledge on water flow and infiltration in mountain permafrost

In steep alpine bedrock, the question of water infiltration and its thermal and mechanical implications is crucial but is rarely addressed directly (Krautblatter et al., 2013b). Studying hydrogeological processes in these environments presents several challenges, including limited accessibility, the hidden nature of water flow pathways, strong spatial and temporal variability, non-linear system behavior, and the difficulty of quantitatively identifying water sources. Hasler et al. (2011) used numerical simulations to explore the impact of advective heat transport by water percolation on subsurface temperatures and ice-level changes. In the absence of hydrogeological field measurements, they performed laboratory experiments and were able to show



significant implications for the role of water flow in thaw-related instabilities in cold mountain permafrost regions. Maréchal
et al. (1999) used a hydrothermal model to simulate an observed thermal anomaly that was found during drilling work in the
road tunnel under the Mont-Blanc massif and showed that infiltration of waters from the surface contributed to the continuous
cooling of the alpine massif at depth. Ben-Asher et al. (2023) estimated the potential water input in steep alpine bedrock using
field measurements and numerical simulations.
Other than indirect studies (Ben-Asher et al., 2023; Hasler et al., 2011; Maréchal et al., 1999; Scherler et al., 2010), very few
studies attempted to directly monitor groundwater flow in steep and high alpine, permafrost-affected environments (Gabrielli
et al., 2012; Manning and Caine, 2007). In a recent study, Scandroglio et al. (2025) measured water outflow in 55 m deep
fractures under the permafrost-affected Zugspitze Ridge (2815–2962 m asl). They compared their dataset with meteorological
data and a snowmelt model to infer hydrological properties and constrain the hydrological pressure in the fractures.
**1.3.  Saturated and non-saturated flow**
From a hydrogeological perspective, a fractured summit is rather characterized as a permeable infiltration zone than as an
aquifer. However, due to the scarcity of drilling data, constraints on the thickness of the high-altitude alpine unsaturated zone
remain highly limited (Maréchal, 1998). The first tests of hydrodynamical models of a high alpine and permafrost-affected
rock wall site were performed by Magnin & Josnin (2021) on the Aiguille du Midi site. This study has shown that the
unsaturated zone is probably more than 1000 m thick. In the Rocky Mountains, several hundred meters of unsaturated zone
above the water table have also been reported (Russell et al. 2001).
Generally, unsaturated conditions apply in soils, permeable rocks, and deposits, and water flow is often considered subvertical
and evaluated using the Richards equation (Smith, 2002). In crystalline rock settings, porosity and permeability are essentially
controlled by the geometry of the fractures network, where most of the water flow takes place. The water flow in fractures is
thus not uniform but occurs along preferential flow paths sometimes called "fingers" (Su et al., 2000), that channel the flow
path at the larger scale of the fractured medium (Tsang et al., 2013). These preferential flow paths are observed both in saturated
and unsaturated fractures (Su et al., 2000).
In the study area of the Mont-Blanc massif, the fracture network opening is highly irregular (millimeters to decimeters
depending on the fractures) and expected to evolve seasonally (Weber et al., 2017). The fractures can also be affected by the
water flow  which can change the extent of ice filling, and develop partially saturated conditions similar to those known from
some epikarsts (Ford and Williams, 1989).
This study is motivated by the need to better understand hydrological processes in high mountain permafrost environments
and connectivity with changing surface conditions, particularly in the context of ongoing climate change. We aim to fill major
knowledge gaps about the timing and quantity of water flow from the surface to the fractures, and how the flow is affected by
conditions at the surface and the source of infiltrating water.



## 2.  Study site and meteorological context

### 2.1.  Aiguille du Midi site

The Aiguille du Midi (AdM) is a peak composed of three granite pillars - Piton Nord, Piton Central, and Piton Sud, with the central pillar (Piton Central) reaching an elevation of 3842 m asl and towering approximately 3000 m above the valley of Chamonix (Figure 1). The site lies on the NW flank of the Mont Blanc massif (MBM) that covers an area of about 550 km² and is oriented NW-SE between France, Italy, and Switzerland.  It is a part of the external crystalline massif of the Alps, whose uplift started about 22 Myr ago (Leloup et al., 2005). It is composed of two main lithological units: a Variscan metamorphic series (mostly gneisses and mafic schists) (453 ± 3 Ma) on the W and S side, and an intrusion of calc-alkaline granite (304 ± 3 Ma) with aplitic veins (Raumer and Bussy, 2004). The MBM is crosscut by a network of shear zones and faults that are mostly oriented N–S and dip sub-vertically (Rossi et al., 2005). Across the NW-SE transect of the massif, these shear zones show a fan geometry, verging NW on the NW part of the massif, and SE on the SE (Bertini et al., 1985), which delineates the main topographical features. High and steep granite rock faces and peaks are typical of the MBM that hosts 28 summits above 4000 m asl, including the roof of the European Alps: the Mont Blanc whose ice cap reached 4805 m asl in 2023.

Glaciers occupied about 100 km² in the late 2000s (Gardent et al., 2014) while permafrost is largely present above roughly 2600 m in N faces and 3200 m in S faces (Magnin et al., 2015a).

The combination of steepness, permafrost, and glacial dynamics results in highly active morphodynamics (Deline et al., 2015). Over the past decades, rockfall ( volume > 100 m³) frequency has significantly increased (Ravanel and Deline, 2011), notably during the hot summers. The main cause was suggested to be permafrost degradation (Ravanel et al., 2017; Legay et al., 2021; Magnin et al., 2023). Permafrost investigation started in mid-2000s in the MBM, with the installation of various temperature sensors in AdM (Magnin et al., 2015b), including 10-m deep boreholes, which registered a temperature increase > 1 °C during the 2011-2020 decade (Magnin et al., 2024).

AdM has been chosen as a pilot site for alpine permafrost investigations because of its representativeness of high alpine rockwalls and its accessibility from Chamonix by a cable-car. Man-made galleries, terraces, bridges, and an elevator allow the visitors to access different parts of the site. Since the hot summer of 2015, water flowing from the fractured gallery walls has become a problem for the operating company (the *Compagnie du Mont Blanc*), leading to the installation of a drained metal plate ceiling to divert the flowing water and keep some parts of the galleries dry for visitors.



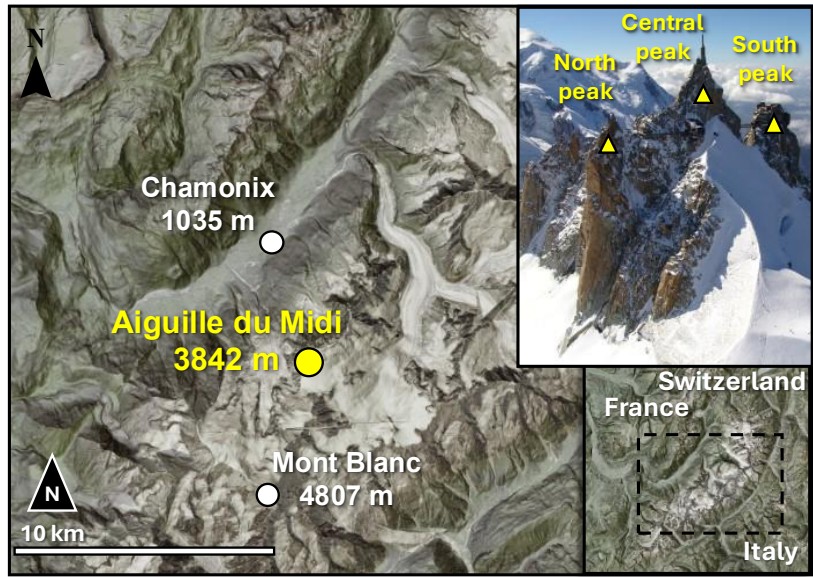

**Figure** 1: **Location of the Aiguille du Midi in the Mont Blanc massif and view of the three peaks at Aiguille du Midi. Picture: S. Gruber. Maps provided by the Federal Office of Topography swisstopo.**

## 2.2. Meteorological conditions in 2022 and 2023

In Europe, the years 2022 and 2023, respectively, were the third and second warmest years in records, after 2020, with a mean annual air temperature (AT) being about 1.1 °C above the 1991-2020 average (ESOTC 2023, 2024). Summer 2022 was the warmest summer ever recorded, outpacing the 1991-2020 average by 1.4°C (ESOTC, 2023). September 2023 was the warmest September on record (EOSTC, 2023).

AT and precipitation records in the town of Chamonix (France), located in the valley just north of AdM, started in the early 20[th] century and are well-suited to characterize the local precipitation regime. At the AdM, AT records started in 2007 but are affected by numerous gaps that sometimes last several months, making this data less reliable for multi-annual comparison.

AT in AdM and precipitation in Chamonix, from 2022 and 2023 are displayed in Figure 2. In 2022, winter and early spring (January-April) were drier (198 and 405 mm recorded at Chamonix over January-April) and warmer (+1.4 °C) than 2023 with an average of -10.9°C and -12.3°C respectively at the AdM. A late spring heat wave led to the warmest May ever recorded in Chamonix with a temperature anomaly of +2.6 °C compared to the 1993-2022 period (continuous hourly records started in 1993 in Chamonix), while May 2023 was average. The mean ATs in June-August were similar for both years. The late summer heat wave in 2023 led to the warmest September ever recorded in Chamonix with a temperature anomaly of +3.1 °C compared to the 1993-2022 period, while September 2022 was average.

Precipitation in spring 2022 was less than half of that in 2023 (184 and 418 mm, respectively, over March-May) while summer 2023 was only slightly wetter than in 2022 (+ 32 mm over June-August). June and September 2022 were wetter than in 2023





and July and August 2023 were wetter than in 2022. Autumn 2023 was also wetter than 2022 (+168 mm over September-
November).
In summary, 2022 was characterized by a very early but long-lasting and record-breaking summer heat wave, while 2023 was
characterized by a late and record-breaking summer heat wave with significantly more precipitation than in 2022.

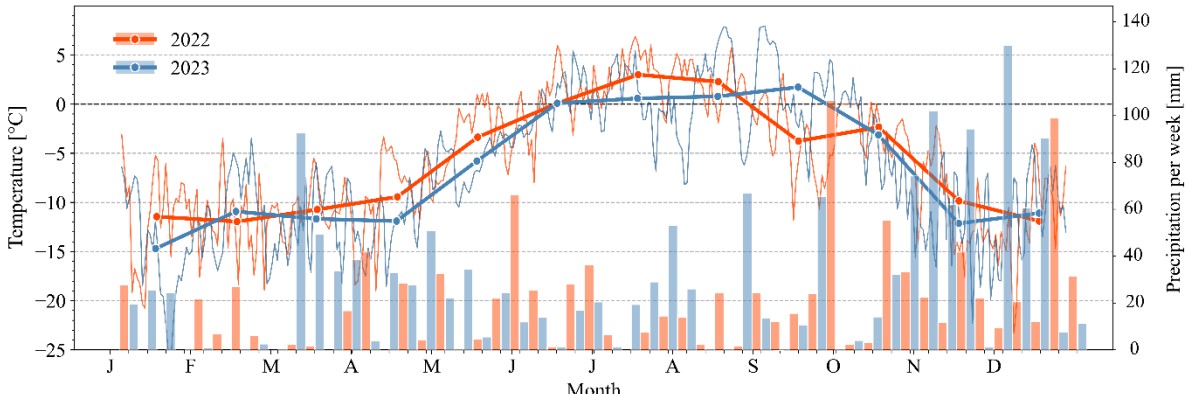


**Figure 2: Daily (thin lines) and monthly (thick lines) air temperature and weekly precipitation in 2022 and 2023 (bars). Air**
**temperature was measured at the top of the Aiguille du Midi and precipitation was measured in Chamonix (1042 m asl). Data**
**provided by Météo France.**

## 3.    Method

In April and May 2022 a monitoring system was installed to measure characteristics of water flowing through fractures in the
roof of the gallery in AdM (Figure 3, Figure 4). Fluorescent dyes were poured into the snowpack on the rock face above the
gallery to trace the water. Ground Surface Temperature (GST) sensors were installed on the rock surface, below and around
the snowpack where fluorescent dyes were inserted. Figure 3 illustrates the entire methodological approach of the monitoring
system.



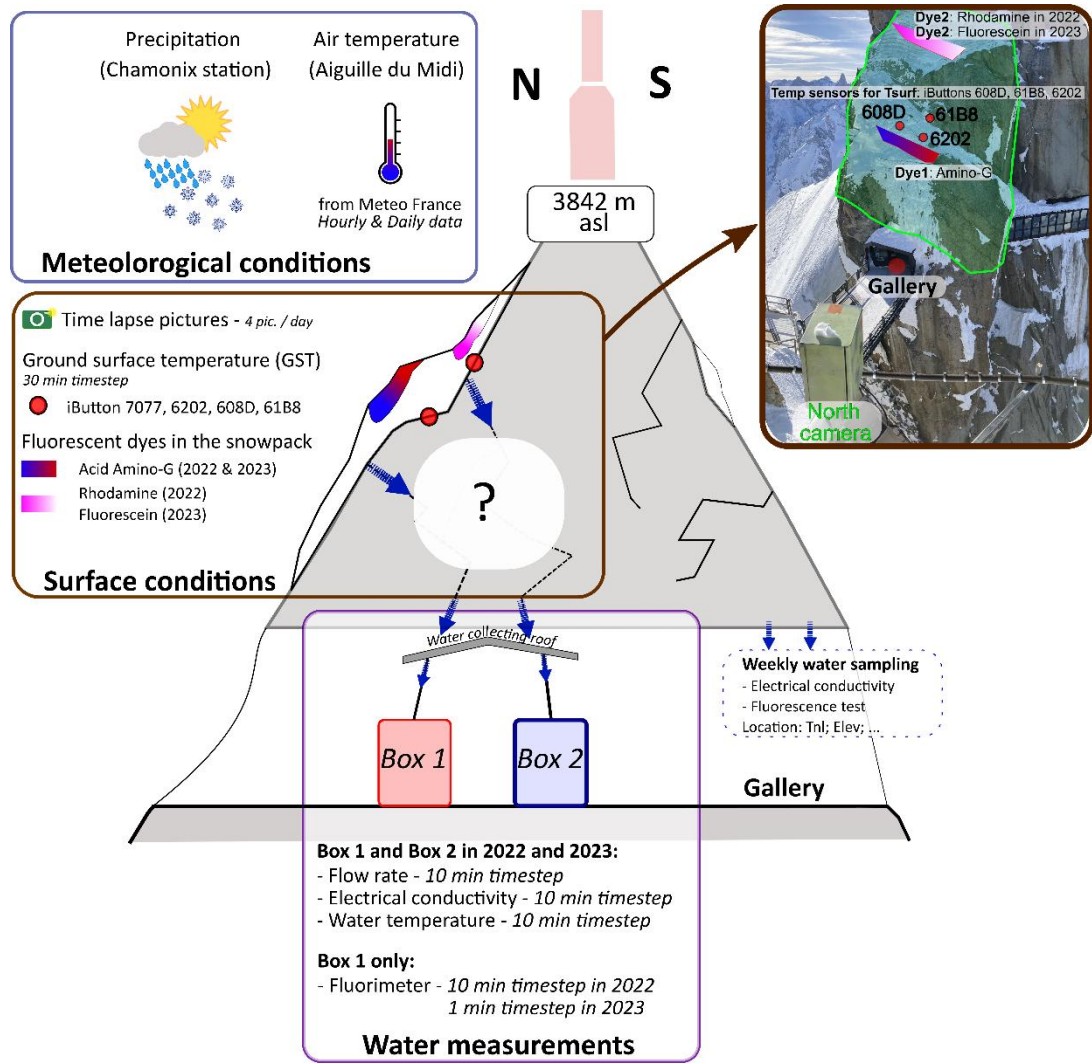

**Figure 3: Sketch of the methodological approach to track and monitor water flows in the Aiguille du Midi central pillar**

### 3.1. Fluorescent dyes in the snowpack

Fluorescent dyes were inserted into the snowpack at the surface above the gallery in two locations, directly above the fractures (5-15 m), to trace the water source. In the 2022 season, two dye solutions were used: 20 L of sulphorhodamine-B solution with a concentration of 0.001 g/L and 20 L of amino acid G with a concentration of 20 g/L. The solutions were prepared and carried in "Ondine®" mineral water bottles with the original mineral water, each with a volume of 5 L. The relatively low concentration of sulphorhodamine-B was chosen to have a light but detectable pink color, far above the detection limit of the fluorimeter sensor.



In 2023, new concentrations were prepared for both tracers, and sulphorhodamine-B was replaced by fluorescein dye, a much
more soluble and detectable dye, to avoid confusion with sulphorhodamine-B from the previous year.
Four bottles of 4 L and six bottles of 1.5 L of "Ondine®" mineral water were used to prepare the tracers solutions. The dyes
in powder have been inserted directly into the bottles with the original mineral water. 9 L of fluorescein solution with a
concentration of 0.667 g/L and 16 L of amino acid G with a concentration of 12.5 g/L
In both years of the study, tracers were injected into the snowpack at the same two locations on the north face of the central
peak (Figure 3). Sulphorhodamine-B in 2022 and Fluorescein in 2023 were injected on the "upper" terrace of the face, while
amino acid G was injected on the "lower" terrace in 2022 and 2023. Tracers were inserted in spring, before the flow started,
on the 11th May 2022 and 22nd March 2023.  We poured the solution in 5-10 points in each terrace and on the snowpack surface.
**3.2.  Ground surface temperature at the snow-rock interface**
Four miniature temperature sensors (iButtons, Mouser®) have been installed in holes drilled 5 cm into the rock surface, at the
snow-rock interface, in the vicinity of the fluorescent dye injection area. The holes with the coin-sized sensors were filled with
gray polymer clay to isolate from direct solar radiation on the metal sensor. They monitored GST at hourly time steps and over
different periods (Table 1), but only one (#6202) monitored the temperatures during both seasons.

| Sensor ID | Monitoring period covered | Location on the face |
|---|---|---|
| *7077* | 11/05/2022 to 23/09/2022 | In drilled hole at base of vertical rock outcrop |
| *6202* | 11/05/2022 to 22/08/2023 | In drilled hole at the surface of the lower terrace |
| *608D* | 06/09/2022 to 22/08/2023 | In a hole drilled at the base of a rock outcrop above the lower terrace |
| *61B8* | 23/09/2022 to 22/08/2023 | In the rock crack (same rock as 608D) |

**Table 1: Miniature temperature sensor characteristics at the snow-rock interface**
Snow melting was detected as "zero-curtain" periods in GST (Figure 5). These periods are characterized by stagnant GST at
~0 °C (Hanson and Hoelzel, 2004). The complete melting of the snow is marked by the transition from dampened GST daily
oscillations to positive and significant daily oscillations marks once the insulating layer of the snow is removed and solar
radiation reaches the rock surface. As GST is measured at point-scale measurements, it lacks spatial representativeness of the
snow melting surface area. Thus, to complete this data, pictures were frequently taken during fieldwork to document snow
patch evolution in 2022, and in 2023, an automatic camera was installed on a terrace of the North Pillar (North Camera on
Figure). It took 4 pictures a day from 1 March 2023 to 22 August 2023, of the north face of the Central Pillar, to monitor the
snow patch evolution right above the water collection system. Pictures after 22 August 2023 are not usable because the
protective glass was broken, and pictures became blurred.



### 3.3. Water flows and temperature monitoring in the gallery

We installed a real-time monitoring system in May 2022 in the west gallery of the Central Pillar, to characterize the water flowing from fractures that cross the gallery. We took advantage of an existing water diversion ceiling set up by the operating company (Figure 4), made of a convex metallic plate collecting water drips and flows, and diverting them to two pipes on each side of the gallery (east and west) to drain water outside. Preliminary observations revealed that water was mostly dripping from two adjacent fracture systems.

Instrumentation included two rain gauges that were installed on each pipe to measure water flow rate (L/h), in protective boxes (Box 1 on the west side and Box 2 on the east side). Water temperature (°C) and electrical conductivity (S/cm) were also monitored with sensors placed on the metallic roof, below the identified water drips. The sensors were plunged in specially designed and 3D printed siphon shaped pipe to maintain a high water exchange rate and minimal water level for detection. As a conductivity benchmark, we measured a value of 9.2 µS/cm from a melted snow sample collected on the 26$^{th}$ of July 2023 and corresponds with known values for snowmelt samples (Brennan et al., 2020; Thompson et al., 2016). In addition, water fluorescence (arbitrary units) was monitored in real-time with a probe (GGUN FL-24) inserted in Box 1 to detect the specific emission spectrum of the dye tracer used: Acid Amino-G and sulphorhodamine-B in 2022 and Amino-acid-G and Fluorescein in 2023 (see Sect. 3.2 for dye spraying strategy). The fluorescence sensor installed in 2022 malfunctioned during the 2022 winter and was replaced by a new probe (STREAM model, TRAQUA®) on the 31$^{st}$ of May 2023, and was removed on the 22$^{nd}$ of August several weeks after the last dye signal was detected.

Every 10 minutes, five measurements were averaged and recorded with a PC400 Campbell Scientific data logger. In addition, data from miniature temperature sensors (iButtons, Mouser®) that were previously installed, were used to monitor the gallery wall (bedrock) temperature.

The site was visited weekly to retrieve data and to clean the rain gauge where sediments were sometimes accumulating (Figure 4).

In 2023 the conductivity sensor was fixed directly below fracture, above box 1, during the thawing season, excluding a short period from 25 July 2023 to 10 August 2023 because of a storm. In addition, electrical conductivity was also measured manually directly in water samples in Box 1, using a mobile sensor, once a week.





**Figure 4: Real-time monitoring system in the gallery, including Box 1 and 2 with the flow gauge (purple frame), siphon with fluorescence probe (red frame). Images of Box 2 with sediments are shown in the green frame.**

### 3.4. Water sampling in the galleries and laboratory analysis

Water samples were collected weekly from Box 1 during both 2022 and 2023 melting seasons. Other locations in the gallery (labeled Box 2, TNL) were occasionally sampled in 2022 and weekly in 2023. Samples were taken in 125 mL brown glass bottles. During the 2023 season, each sample was measured for electrical conductivity, directly after collection. The bottles were stored in a fridge, protected from light to minimize biological activity.



Further high-resolution fluorescence analysis of the water samples was carried out in a laboratory, using a fluorescence
spectrophotometer (Varian Cary Eclipse) to validate the real-time fluorimeter data. The samples were exposed to light with
the characteristic wavelength spectrum matching the excitation spectrum of the dye tracers used in the experiment. The
emission vs. excitation wavelength plots were used to find peaks in emission distribution that corresponded to the presence of
the dye tracers.
In addition to fluorescence analysis, we performed stable isotope analysis on 11 water samples to determine δ¹⁸O and δD
values. These stable isotopes are widely used in hydrological studies to trace the origin and history of water, as their ratios are
sensitive to fractionation during phase changes in the hydrologic cycle. Such analyses can reveal important information about
water sources (e.g. snowmelt vs. rainfall), transport pathways, and storage times. By comparing the measured isotopic
signatures to the Global Meteoric Water Line (GMWL), we can assess whether the water follows typical meteoric patterns or
has undergone secondary processes such as evaporation, mixing, or prolonged subsurface residence. Deviations from the
GMWL can also indicate elevation effects or seasonal variations in precipitation, making isotope data a valuable complement
to physical and chemical tracers in characterizing alpine hydrological systems.
### 3.5. Data analysis
3.6. We processed and analyzed the continuous time series data by developing codes in Python3 and MATLAB. All the
235        time series were filtered for erroneous values and interpolated to evenly spaced time steps for consistency.

### 3.6.1. Recession curves analysis
Recession curves have been studied since the late 19th century (Brutsaert and Nieber, 1977; Tallaksen, 1995) and are
commonly used in hydrology to interpret the flow behavior and characteristics of aquifers. A key advantage of this approach
is that it allows the derivation of empirical, quantitative parameters that reflect the subsurface drainage. Following work by
Boussinesq (1877), Maillet (1905) suggested an exponential analytical solution to describe aquifer drainage behavior:
$$Q(t) = Q_0 e^{-at} \qquad (1)$$
where $Q$ is flow rate, $t$ is time, $Q_0$ is peak flow rate, and $\alpha$ is the recession coefficient. To account for flood recession in a
channelized flow, we opted for the general form suggested by Brutsaert and Nieber (1977) that is commonly used for river
flood recessions (Brutsaert and Nieber, 1977; Krakauer and Temimi, 2011):
$$\frac{dQ}{dt} = -aQ^b \qquad (2)$$
which can be integrated and solved for $Q(t)$ as:
$$Q(t) = (Q_0^{1-b} - a(1-b)t)^{\frac{1}{1-b}} \qquad (3)$$



where *a* and *b* are constant coefficients. Note that the integration of Equation 2 in the case of *b=1* corresponds to the form of
exponential decay as expressed by Equation 1. Scandroglio et al. (2025) recently applied Maillet's law (Equation 1) to analyze
flow in fractures within a permafrost-affected rock wall in the Northern Calcareous Alps, at the German-Austrian border. Their
study focused on a 55 m deep tunnel in karst limestones, where flow paths could extend for several hundred meters. In contrast,
in our study, flow is confined to widely open, sub-vertical granite fractures with a maximum path length of 30 meters.
Additionally, Scandroglio et al. (2025) used a single best-fit curve for their entire dataset, which included only 23 events over
eight years. In comparison, we used 93 events for recession curve analysis (out of 144 events that were recorded, see 4.1.5 for
more details) over two consecutive. To capture temporal variations in flow characteristics, we developed an algorithm that
automatically fits a separate curve to individual events, allowing us to track changes over time and compare flow behavior
between Box 1 and Box 2 (Figure S2).
**3.6.2. Moving window cross-correlation**
A moving window cross-correlation analysis was used to estimate correlations and lag times between measurement time series.
All the time series were filtered with a high-pass moving-average filter of one hour to reduce noise. Cross-correlation analysis
was carried out on days when the maximum flow rate exceeded 6 L/h. This value was chosen as a threshold to filter data from
days with well-defined hydrograph curves. These include data from 109. The analysis was made on the entire period of
observed water flow in each season, i.e. mid-May to August in 2022 and June to September in 2023.
**4.  Results**
**4.1.  Water flow rate**
**4.1.1. Seasonal pattern**
Water flow is highly seasonal. In both years, water mostly flowed between May and October (Figure 5), with periods of
sporadic and continuous flow that can last several weeks. The occurrence of water flows correlates with the occurrence of
positive AT (Figure 5).
Sustained periods of water flow were mainly observed from late May to mid-September in 2022, and from mid-June to late
September in 2023. The timing and magnitude of flow differed between Box 1 and Box 2 (**Error! Reference source not
found.**). In both years, water flow in Box 1 started several weeks before that in Box 2. In general, the amount of water in Box
2 increased through the summer season.



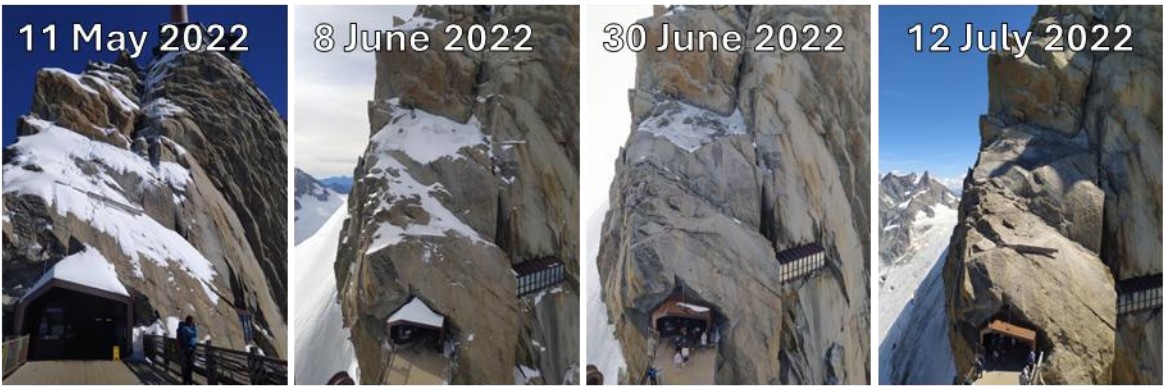

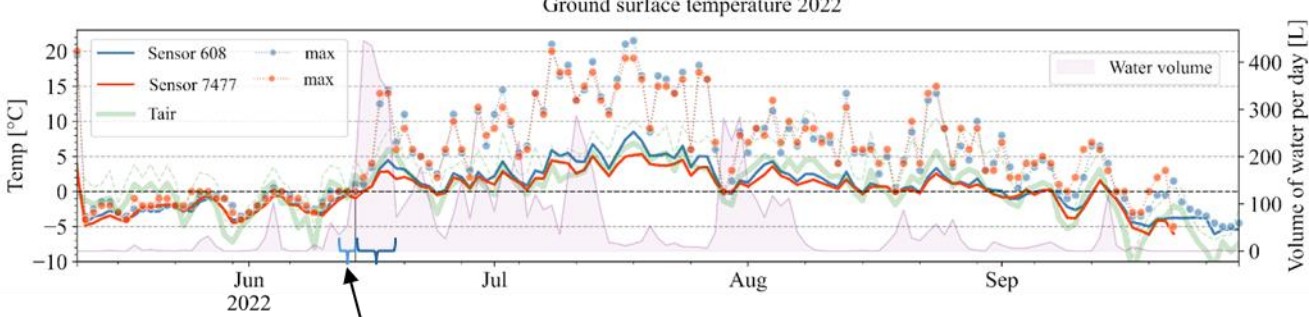

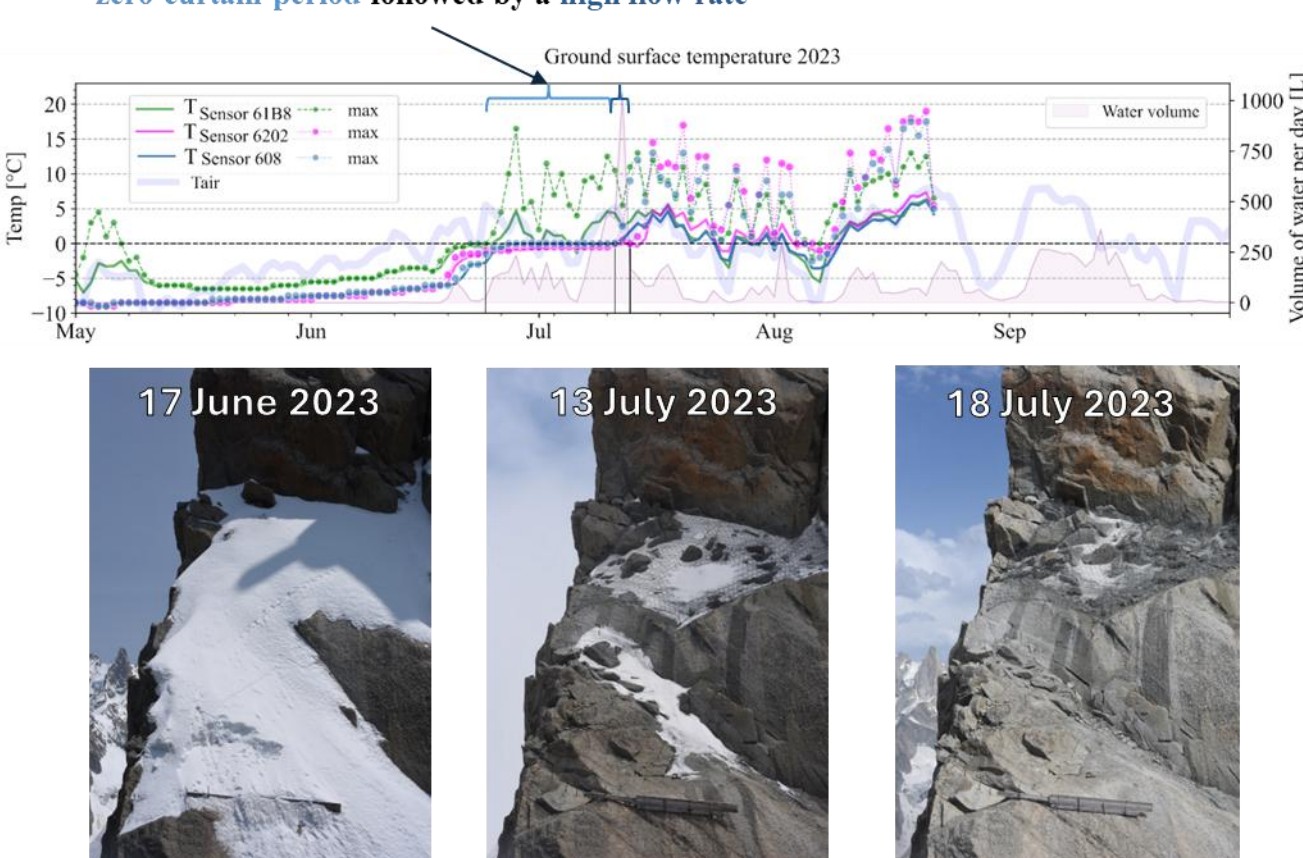





**Figure 5: GST are measured on the NE face, above the gallery entrance, directly above the monitoring system. Flow rate is measured at output from rock fractures in the gallery wall. Solid lines represent the daily mean. Note the zero curtain period which marks the thawing of the snowpack and exposure of the rock surface to atmospheric heating. Photos showing the evolution of the snow cover during the thawing seasons in 2022 and 2023**

**In 2022**, the total volume of water flowing through the monitoring system (Box 1 and Box 2) was 8001 L (**Error! Reference source not found.**). About 70% of this total volume (5621 L) was collected in Box 1, while about 30% reached Box 2 (2380 L). 75% of the total volume in Box 1 occurred between the 11th of June and the 14th of July (4216 L). From the total flow volume in Box 2, 74% flowed in two relatively short periods: between the 14th-19th of June (496 L) and between the 28th of July and the 8th of August (1257 L).

**In 2023**, the total volume of water flow was 11605 L – 45% more than in 2022, of which 61% (7079 L) flowed through Box 1, and 39% (4526 L) in Box 2. 75% of the total volume in Box 1 occurred between the 19th of June and the 10th of August (5309 L). In Box 2, almost the entire volume (95%) flowed in three relatively short periods (3 to 18 days): between the 8th-13th of July (831 L), between the 22nd-25th of August (611 L) and between the 1st-18th of September (2851 L).

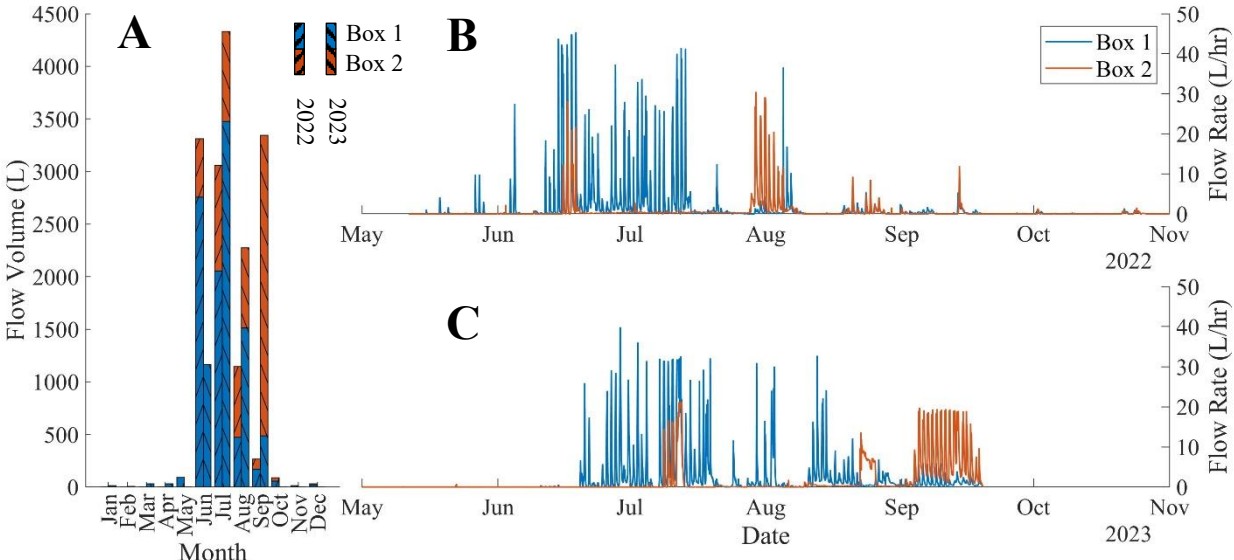

**Figure 6: A) Monthly distribution of flow volume in Box 1 and Box 2 during the 2022 and 2023 seasons. B) Measured flow rate vs. time in 2022. C) Measured flow rate vs. time in 2023.**





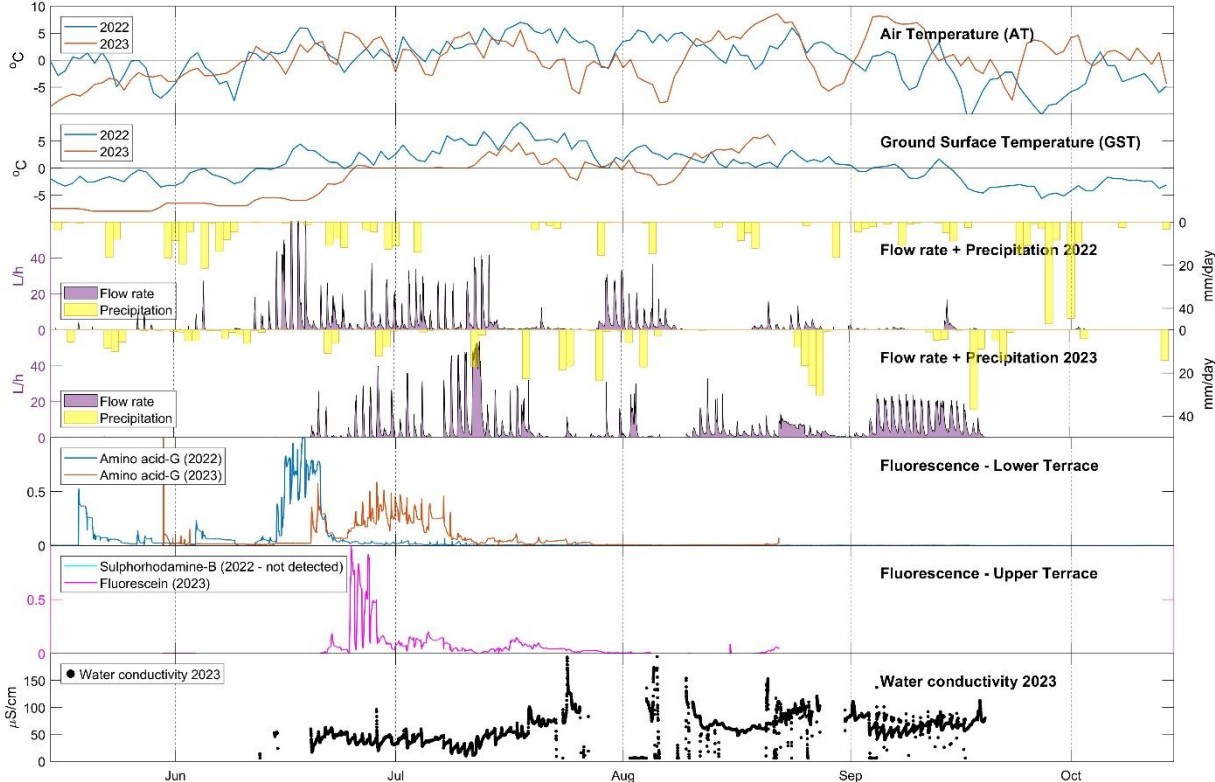

**Figure 7: Annual time series**

### 4.1.2. Daily patterns and flow rates

The observed flow rate presents daily cycles (Figure 7) with peak flow rates generally occurring between 17:00 to 20:00 (Table 2, Figure S1), and minimum flow rates two orders of magnitude lower during the morning time.

| | Average time of signal peak | | Av. time difference with Flow Rate signal | |
|---|---|---|---|---|
| | **2022** | **2023** | **2022** | **2023** |
| *Air temperature (AT)* | 11:40 | 11:30 | - 7 h | - 6 h 50 min |
| *Ground surface temperature (GST)* | 15:00 | 14:00 | - 3 h 40 min | - 4 h 20 min |
| *Flow rate* | 18:40 | 18:20 | -- | -- |

**Table 2: Average timing and lag time of flow peak and AT.**

Results of a moving window cross-correlation show that daily flow rate oscillations are correlated with AT measurements with a lag time of 3-9 hours, and with GST with a lag time of 0-3 hours. This lag time was found to be consistent in both years of the experiment (Table 2, Figure S3).



### 4.1.3. 2022 season

In 2022, the first flow event was recorded in the system in Box 1 on 15 May 2022 around 17:00 and reached a maximum flux of 1.1 L/h (Figure 7). Sporadic daily events continued to occur in Box 1 with a general trend of increasing flow rate (reaching 9.9 L/h) until the 11th of June. From that date, a 5-week period of continuous flow started in concomitance with positive AT, and daily cycles sometimes reaching rates of > 40 L/h. The continuous water flow period ended on the 14 July 2022, although the AT was still positive and even reached its maximum value, likely indicating the complete thawing of the winter snowpack. The water flow events that followed and recorded in Box 1 are linked to precipitation that most likely fell as rain since the AT was positive (Figure 5, Figure 7). In Box 2, the first flow event occurred early in June, with the first significant event (> 10 L/h) on 15  June 2022, when the AT increased, and after a short period of low flow rates. This event was followed by 5 days of continuous flow with daily oscillations reaching > 25 L/h, similar to those observed in Box 1 at that period, but starting 5 days later (Figure 7). A period of low flow rates continued in Box 2 until the 28th of July when an 8-day period of high daily flow rate reaching up to 30 L/h with strong daily oscillations started with a precipitation episode, two weeks after the end of the high flow rate period in Box 1. A few low flow rate events still occurred between October and December.

The maximum flow rate recorded was 45.4 L/h in Box 1 (18th of June 2022 at 18:40), and 30.46 L/h in Box 2 (29th of July 2022 at 18:50). Cumulating flows from Box 1 and Box 2, the maximum flow rate was 67.35 L/h with 39.42 L/h in Box 1 and 27.96 L/h in Box 2 the 16th of June 2022 at 19:20. The maximum daily amount was 445.9 L on the 15th of June 2022.

### 4.1.4. 2023 season

In 2023, the first flow started with a significant oscillation on 19 June 2023 in Box 1, that was the onset of continuous daily oscillations, with an increasing trend in maximum daily flow rate reaching 40 L/h on 28 June 2023. The flow rate then decreased with to maximum daily values < 10 L/h, and a whole day without flow occurred on the 6 July 2023. This interruption in daily oscillations coincides with a short-term decrease of AT below 0°C. From the 20 July 2023 to the 10 August 2023, daily oscillations were more sporadic, in lagged correlation with AT oscillations, water flow occurring under positive AT and no flow under negative AT. The largest daily oscillations during this period, sometimes reaching up to 30 L/h, are linked to precipitation episodes with ATs sometimes close to 0 °C, hinting at possible rain events (Figure 5, Figure 7). From the 11 to 22 August 2023, continuous daily oscillations occurred again as the AT reached its maximum of the year and were stopped by a decrease in AT coinciding with an intense precipitation episode (reaching up to > 35 mm/d). A final flow phase occurred in Box 1 in September, with regular daily oscillations but rather low rates (around 5 L/h).

Box 2 experienced its first daily oscillations between the 8th and 13th of July, after about 3 weeks of generally positive AT, but with maximum daily oscillations remaining below those of Box 1 (up to 20 L/h against 30 L/h). A second water flow episode occurred between the 22nd and the 25th of August. This event was unusual because it lasted for 3 days without the characteristic daily recession but continued steadily at a flow rate of 5-12 L/h between the morning of the 22nd to the night of the 25th of August. This event is also enigmatic because it occurred after two weeks without precipitation documented in Chamonix and





relatively high temperatures, during which Box 1 flowed regularly. It was followed by a 9-day period without flow. Final flow
phase with regular daily oscillations occurred between the 3$^{rd}$ and 19$^{th}$ of September, and with maximum flow rates of 20 L/h,
that is 4 times higher than in Box 1 during the same period. This important water flow episode occurred with the onset of
positive AT, following a several days period of temperature drop below 0 °C and precipitation episode that likely occurred as
snow.
There was also an exceptional one-off event between 11 and 13 July 2023, when, on both boxes, the flow continued for 2 full
days after a precipitation event and during a short-term decrease in AT (despite still being positive).
In 2023, the maximum flow rate observed separately was 39.83 L/h in Box 1 on 28 June 2023 at 19:40, and 21.20 L/h in Box
2 on 12 July 2023 at 15:10. Cumulating flows from Box 1 and Box 2, the maximum flow rate observed was 53.81 L/h with
32.58 L/h in Box 1 and 21.18 L/h in Box 2 the 12 July 2023 at 15:10. The daily maximum volume was 1032.99 L on 1 July

341    2023.

**4.1.5. Recession analysis**
The exponential recession curves (Eq.1) fit well with the observed daily events, with an average $R^2$ value of 0.93. Values with
$R^2$ values below 0.8 were omitted from the analysis, resulting in 93 events (Figure 8). The 'a' coefficient shows a clear
decreasing trend in time from values of 9-10 to 5-8 in both 2022 and 2023 seasons (Figure 8), while the 'b' coefficient increases
from values of b≈1 at the beginning of the melting season to values of b≈1.15 at the end of the season (Equations 2 and 3).





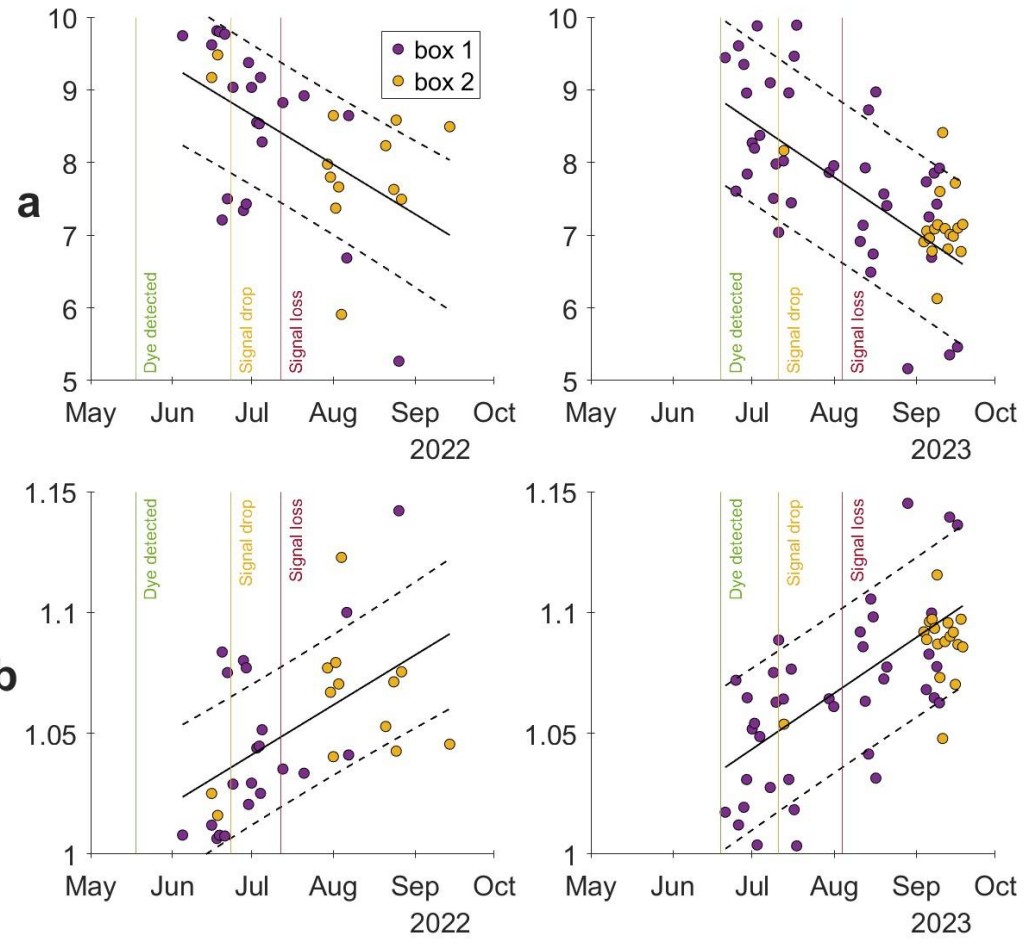

**Figure 8: Values of the recession curve exponential coefficients a (top) and b (bottom) in 2022 (left) and 2023 (right) vs. time. Values obtained from curves with R² values below 0.75 were omitted from the analysis. The black line is the linear regression of all the points (box 1 + box 2) with ±standard error (dashed black lines). The vertical lines indicate the timing of the detection of the fluorescent dye in the water that exits the fractures (green), the rapid drop of the signal intensity (orange), and the disappearance of the signal (red).**

## 4.2. Snowpack evolution and water flow characteristics

Snowpack evolution is assessed through GST measurements at the snow-rock interface and using time-lapse pictures in 2023. Dampened daily oscillations in GST are interpreted as the presence of a snowpack with sufficient insulating effect. The melting period is generally visible as a zero-curtain period (*i.e* persisting 0 °C conditions at the rock-snow interface) (Hanson and Hoelzle, 2004) that lasts several days to several weeks.



Measured GST data at the rock-snow interface are displayed in **Error! Reference source not found.** In 2022, the dampened
daily oscillations are revealed by the similar values of mean and maximum GST, and the zero-curtain period is nearly
nonexistent. This could be related to the early heat wave in 2022 that accelerated snow melting. Nonetheless, the first water
flow events in May and early June 2022 occurred with GST rising close to 0 °C, hinting at a link with snow melting. The first
significant water flow event in 2022, which is also the greatest one, with values reaching > 400 L/day coincides with the
transition to positive GST around mid-June. Summer precipitation episodes are suggested when water flow events follow
periods with limited water flows and high GST, such as in late July 2022. In 2023, the effect of snow cover on GST patterns
is more evident, with an initial period of non-existent daily oscillations followed by a zero-curtain period until mid-July. First
flow events occurred during the onset of the snow melting period, with the highest peak of water flow reaching > 1000 L/day
at the end of the *zero-curtain period*.

369        **4.3.  Fluorescence**

370        **4.3.1. Real-time fluorescence monitoring**

In 2022, the real-time fluorescence sensor shows a strong signal of amino acid-G that followed the very first flow events in
mid-May 2022 (Figure 7) and the sporadic flow events that followed it until 11 June 2022. The high amino acid-G signal
continued with the onset of continuous water flows around mid-June 2022, until the rapid decrease at the end of June. The
disappearance of the fluorescent signal, despite the sustained water flow likely corresponds to the complete melting of the
lower terrace snowpack that contained the amino-G tracer, as seen in the photos from the time lapse camera (Figure 5). A weak
signal of amino acid-G was detected until the middle of July 2022 when both the fluorescence and flow rates diminished. The
period of weak amino acid G signal in the water could indicate dilution with water from precipitation that occurred after the
dye was inserted or another not-dyed source (either late snow or rain).
No signal of the sulphorhodamine-B tracer inserted in the upper ledge was found. This could be due to excess dilution of the
tracer solution with the snowmelt water to concentrations that were below the sensor sensitivity. A second hypothesis is that
snowmelt from the upper terrace did not reach the fracture. In 2023, a different dye was used (fluorescein), with a significantly
higher concentration (see Sect. 3.1), and was clearly detected, suggesting that the absence of sulphorhodamine-B signal in
2022 was due to dilution.
In 2023, the amino acid-G signal was also detected in the first water flows on 19 June 2023. From the 24 June 2023 onwards,
the signal of the fluorescein dye, was detected alongside amino acid-G, likely confirming that the concentration in
sulphorhodamine-B was probably too low in 2022. The fluorescein signal is shorter than the amino acid-G signal, with a single
high peak in the last week of June followed by a rapid decrease to low values. This could be explained by the different pathways
from the upper terrace snowpack through the fracture network, together with the effect of low dispersivity of fluorescein. The
amino acid-G high peaks persisted continuously until the 8[th] of July. Afterward, both tracers remained in low concentrations





until the end of July, suggesting that much of the snow that had fallen during the winter and spring was removed by the 27th
of July. That could mean that all the winter snowpack infiltration time is a little bit more than one month. Thereafter, the water
that flowed after the mid-end of July was either direct precipitation (rain and snow) or possibly melting of ice that predated
the insertion of the tracers.
### 4.3.2.    Fluorescence laboratory results
Additional analyses were carried out in the laboratory on water samples collected between May and mid-July 2023 from Boxes
1 and 2 and various fractures dripping into the galleries of AdM.
Analyses done with a high sensitivity spectrophotometer in the EDYTEM laboratory confirm the results found from the
TRAQUA sensor in Box 1. The signals for Fluorescein and amino-G show peaks at the same periods, i.e. at the end of the
month of June and beginning of July. The fluorescein signal is very short-lived, unlike amino-G, which continues to appear
for a longer period. Samples from other locations in the gallery did not show a signal of any of the fluorescent dyes.
### 4.4. Stable isotopes
Analysis of oxygen and hydrogen isotopes in the water samples shows that $\delta^{18}O$ and $\delta D$ values range between -3.2 to -10 and
-15 to -73 respectively (Figure 9). These values are consistent with those reported in high-elevation mountain regions, such as
the Alps (Lauber and Goldscheider, 2014), the Pyrenees (Herms et al., 2019), and Northern China mountains (Sun et al., 2016).
Excluding two samples, the $\delta^{18}O/\delta D$ ratio in all the water samples fall close to the global meteoric water line (GMWL) or align
on a straight line parallel to the GMWL, likely because of a seasonal evolution of the local meteoric line from the GMWL (i.e.
three samples taken on the 22nd of September -labeled _22SEP). Two samples taken on the 28th July 2022 from Box 2 deviate
significantly below the GMWL (BOX2_28JUL – directly from the fracture, BCKT2_28JUL – from a 5L bucket that collected
the water from the fracture) (Figure 9). This suggests that the water emerging from the fracture above Box 2 on that day was
not of recent meteoric origin. A possible explanation is an extended residence time within the fracture system, allowing for
interactions with the surrounding rock. Water samples collected during the peak of summer, on July 28 (BOX2_28JUL,
BCKT2_28JUL) and August 9 (BOX1_9AUG, BCKT2_9AUG), from both collection systems (Box 1 and Box 2) exhibit
relatively enriched $\delta^{18}O$ and $\delta D$ values compared to those taken in early summer and fall. This enrichment may indicate the
partial melting of seasonal snow. In contrast, two other samples from different locations within the AdM galleries (TNL_9AUG
and BRCK_28JUL), collected on the same dates, do not show this enrichment. The observed $\delta^{18}O$ and $\delta D$ enrichment during
summer is consistent with findings from the Alps (Lauber and Goldscheider, 2014; Novel, 1995).



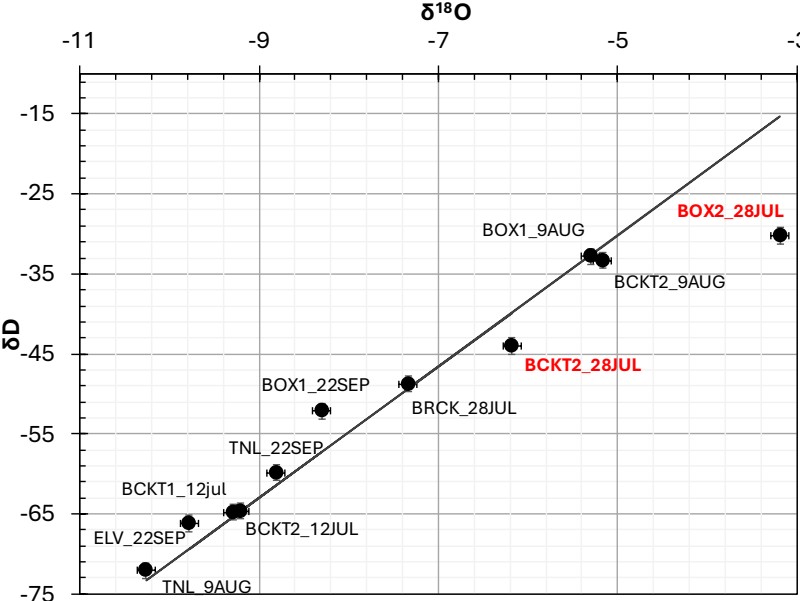


**Figure 9: Stable isotopes δ¹⁸O and δD in water samples. Note the two outliers (labeled in red) from the global meteoric water line**
**(GMWL, black line) in samples taken from Box 2 on the 28ᵗʰ of July 2022.**


## 4.5. Water electrical conductivity


Electrical conductivity values are provided as maximum values per day of flow, corrected at 25°C.
The results of electrical conductivity from the 2022 season were unreliable due to erroneous installation of the sensor. Only
results from 2023 are thus presented and analyzed (Figure 7). Overall, conductivity values were far above the benchmark value
measured in melted snow samples (9.2 µS/cm). On the continuous measurement (real-time monitoring system), the electrical
conductivity of the water flowing into Box 1 was relatively constant from mid-June to mid-July, with daily oscillations between
10-55 µS/cm and a general decreasing trend. The daily oscillations correlate with flow rate in a reverse relation – when flow
rate is high the conductivity decreases (Figure 7). These values of conductivity correspond to the period of continuous cyclic
flow rate in Box 1 that ended with the complete thaw of the winter snowpack). Interestingly, significantly higher conductivities
were measured at other locations in the galleries, such as to 485 µS/cm in a tunnel wall from mid-July onwards, and 430 µS/cm
(measured in 2022) at another location along the "brick wall" near the Hellbronner terrace.

## 4.6. Water temperature


The temperature of the water during flow events measured in two locations at outputs from the fractures ranges between 0 °C
and 13 °C with an average of 6.1 °C (Figure 10). Measurements made during periods without flow or subzero temperatures
were removed from the analysis. Average GST and ATs during the thawing season (15 May to 15 September) are close to



0 °C. Measurements taken at the rock surface in the gallery walls, near the fractures, during flow events show an intermediate
mean value of 3.0 °C.

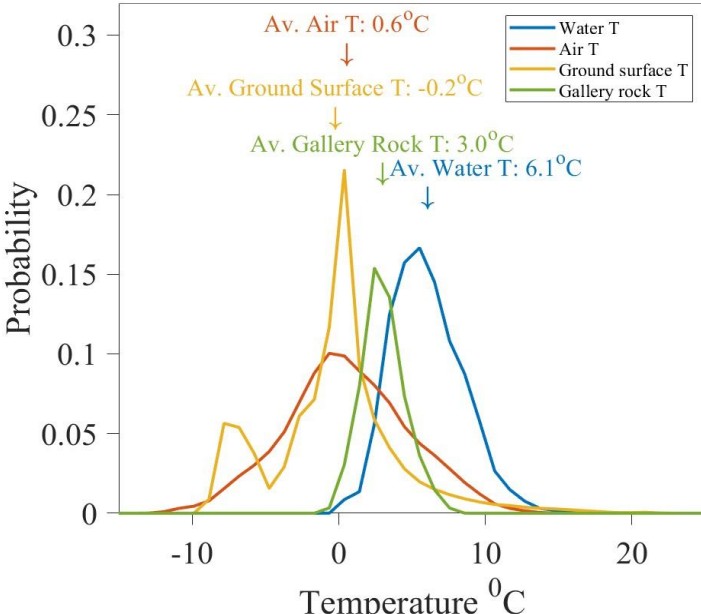

**Figure 10: Distribution of temperatures monitored during flow events (blue), atmospheric ATs (orange), ground surface**
**temperatures (yellow), and gallery wall (green). All distributions show data from the thawing season in 2022 and 2023 (15 May – 15**
**September). Note that the water T distribution (blue) shows only data when water flow was detected in the monitoring system, while**
**the other T distributions represent the entire data within the thawing season.**

## 5.  Discussion

### 5.1.  Water flows and weather conditions

Our results show rare evidence of highly effective surface-subsurface connectivity in steep permafrost-affected slopes, and
strong climatic signals in both seasonal and diurnal scales. We see a clear link between the timing of ATs and GST becoming
positive in the early summer months and the onset of water flow. The first flow events in the season, which appeared in early
May (2022) and June (2023), display a clear signal of the dye tracer and are directly linked to snow melting occurring under
positive AT in the relatively shaded north-exposed rock face.
In both years, the onset of water flow in the fractures occurred when the daytime ATs reached values above 0 °C. This change
in temperature to positive values directly induced the melting of the snow that was deposited during winter, and infiltration
into the fractures. The melting of the snowpack is demonstrated by the simultaneous detection of the fluorescent dye tracers
injected into the snowpack and the zero-curtain effect observed in the GST (Figure 5, Figure 7). The melting of the snowpack
accelerated when the rock surface was exposed to heat from the atmosphere and temperatures turned positive. From this point,





water flow behavior became more uniform, with regular daily oscillations, and reached the highest flow rates. Subsequently,
after the exposure of the rock surface, some of the water in the fracture was from direct precipitation, which likely melted
rapidly on the rock surface as the temperature increased, often above 0 °C. Each year, water flow in the fractures ceased when
the temperature became negative again during the fall months, and icicles appeared in the fractures.
Based on the 2-year monitoring, we conclude that water flow processes in high mountain rock faces are therefore seasonal,
directly linked to the change of air and surface temperatures above 0 °C during the summer period and below 0 °C during fall.
The continuous detection of the dye, together with an analysis of time-lapse photos of the rock face and the shift of GST to
positive values, show that snowmelt is the main source of water in the fractures during the early and main stages of flow, and
contributes most of the water.
GST and AT also control flow rate oscillations on a daily time scale. The rapid increase observed in flow rate in the afternoon,
followed by peak rate in the evening, and a long recession to significantly low flow in the early morning is correlated with the
peak of air AT and GST with a lag time of 3-9 and 0-3 hour respectively (Table 2, Figure S3). The high sensitivity of water
flowing in the fractures to surface temperatures is also demonstrated on July 6[th] 2023, when a single day with temperatures
drop below 0 °C leads to a complete stop in water flow. The observed lag time enables estimation of the water travel time
through the fracture system and allows a rough approximation of flow velocity on the order of ~10 m/hr.
The acceleration in flow rate coincides with the heating of the rock surface to >0 °C and the possible top-down thawing of the
active layer (i.e. the near-surface layer that freezes and thaws through summer).
Nevertheless, the time lag between surface signal and water flow as well as the thawing of the active layer must be cautiously
considered as it is possibly influenced by the open-system of the gallery causing an open flow path and a thermal shortcut
allowing for bottom-up heat transfer.

### 5.1.1. Heat waves effect

The contrast in summer conditions between 2022 and 2023 further illustrates that climatic conditions strongly influence the
timing and characteristics of the water flow period (Sect. 2.2). This is well demonstrated in the effect of the early heat wave in
spring 2022 that resulted in an early onset of water flows and the late heat wave in autumn 2023 that extended the water flow
period much later in the season. The 2023 season was significantly wetter in terms of precipitation, and subsequently, more
water flowed in the monitored fractures. However, when comparing the monthly distribution of flow during the thawing
season, we find that it was greatly influenced by the heat waves. Between May to mid-July, flow volume in 2022 was much
higher than in the same period in 2023, as a result of the early and rapid thawing. Only in late August, did the total volume of
water flow surpass that of 2022 (**Error! Reference source not found.**). This raises an interesting point for future research on
the influence of early vs. late water infiltration in permafrost rocks and the impact on hillslope processes. Assuming that water





that infiltrates later in the season is warmer and the infiltration paths contain less ice, it can potentially accelerate permafrost
degradation and deepening of the active layer.
The rapid thawing in 2022, compared with the thawing in "normal" temperatures in the spring of 2023 is demonstrated in the
GST data and the lack of a zero-curtain period that is seen in the 2023 data (Figure 7). The zero-curtain likely represents a
thawing period in which the snow absorbs latent heat from the atmosphere during the phase transition. We suggest that during
the early heatwave in 2022, the thawing was very rapid, and the latent heat was absorbed rapidly. This can be seen in Figure
5, which shows a large volume of water immediately after GST turns positive in mid-June 2022.

## 5.2. Water flow path conditions

Our results also demonstrate that an effective pathway is available in the fractures network for the infiltration of water from
snowpack melting at the end of spring. The presence of fluorescent dye in the first water flow shows that the transfer rate is
high. Furthermore, it seems that when the water flow stops at the end of fall, and icicles form in the fracture outlet, the flow is
unsaturated. This unsaturated flow shows that there are preferential flow paths into the fractures, leading to open paths available
for the melting water of the snowpack in the following spring. However, here again, we cannot overrule the possibility that
the man-made space of the gallery contributed to the unsaturated conditions. In natural conditions, if undrained conditions
occur, water could accumulate in the fractures, refreeze inside, and seal them. From a geomorphological perspective, artificial
prevention of ice accumulation can inhibit fracture development through ice segregation and the related cryostatic pressure
(Draebing et al., 2014; Draebing and Krautblatter, 2019; Hales and Roering, 2007; Hallet et al., 1991; Matsuoka and Murton,
2008; Matsuoka and Sakai, 1999). Completing these water flow observations with crack-meters measuring fractures rheology
would provide an interesting perspective to clarify the role of the gallery, but this would require well-identifying the fractures
that are directly connected to the gallery.
Interestingly, our monitoring system shows different but consistent timing of water flows in Box 1 and in Box 2, although they
are located only a few meters away (**Error! Reference source not found.**). One reason for the delayed flow in Box 2 could
be linked to the location of the draining area closer to the colder north face, while the draining area of Box 1 is closer to the
west face, which is exposed to more solar radiation. Another explanation could be suggested based on the observed
accumulation of sediments in Box 2, which was not observed in Box 1. The origin of the sand-size sediments observed in Box
2 is very likely from the erosion of the granite rock. That would suggest that the fracture system that is drained to Box 2 is
filled with sediments that reduce the hydraulic conductivity. However, it seems that after the onset of flow in Box 2, it directly
responds to precipitation and positive AT and reaches high flow rates, like those measured in Box 1 (e.g. in September 2023,
**Error! Reference source not found.**). This supports the first explanation of different exposure to solar radiation. The effect
of the sediments filling on the hydraulic conductivity is thus reduced in late summer, perhaps by the thawing of ice-filled pores
in the sediments filling. Alternatively, the sand and ice in the sub-vertical fractures might act as a partial plug which



accumulates water above the infill. When the hydraulic head is high and the ice filling is thawing, the plug can break, the sand
is transported, and the flow regime changes (as it was observed in box 2).
Based on the delayed onset of flow into Box 2 and the different flow behavior when compared with Box 1, we suggest that the
two boxes collect two different flow pathways with some common parts. Because Box 1 and Box 2 are located close to one
another (~3 m long distance), we suggest that under the north face, the fracture network is complex; Some parts of the network
contain sand-size sediments, explaining probably the late and lower flow of Box 2, while other parts are without any sand
filling and have a different hydraulic behavior. The effect of the sediment infill on the fracture hydrology should be investigated
in further work.

### 524  5.3.  Deciphering possible water sources

According to the fluorescence data and the water flow timing, much of the collected water is directly originated from recent
snowmelt. This is also supported by the analysis of stable isotopes in water samples from Box 1 (Figure 9). It is also very
likely that direct rain in late summer infiltrates the rock fractures. However, some data also hints at other possible sources of
water. First, even the lowest values of electric conductivity of the water are far above the expected snow melt conductivity and
they steadily rise with decreasing flow rate, as often observed in aquifers. The surprisingly high electric conductivity found in
some samples collected from fractures can point to long residence times in the rock. Considering the results from the
fluorescent dyes and hydrological behavior, the flow path is very short (in distance and in time), thus ruling out a long exchange
time between the surface water and the rock. An alternative explanation is that recent meteoric water is mixed with older water
trapped as ice in the permafrost-affected rocks. This is also supported by the observed change in the form of the recession
curves with time (Figure 8). The recession curves at the beginning of the melting season (May-June) show values of b≈1 and
fit well with the exponential form that is expressed in Equation 1. The early-season recession curves are also characterized by
high values of 'a'. Over time, the value of 'b' increases linearly to a form better described by Equation 3, while the value of
the 'a' coefficient decreases. This change in recession form, from aquifer-type (Equation 1) to channel-type (Equation 3) can
be explained by the thawing of ice in wide sub-vertical fractures that are likely to react more individually (rather than as a
network) and enable rapid flow in the fractured granite. The decrease of 'a' is non-trivial since one could expect that the
drainage would be more efficient and with shorter recession time (i.e. higher 'a' values) as the thawing of ice in the rock
fractures progresses. We thus suggest that the observed decrease in 'a' is due to a gradual change in the water source. As less
water drains from the surface (after the complete thawing of the winter snow) and more water drains from the subsurface ice
trapped in the fractures, the hydraulic gradient is reduced, and the duration of the recession is extended. Another evidence of
a possible fossilized source is in samples collected from Box 2 on the 28th of July (BOX2_28JUL, BCKT2_28JUL) which
shows an isotopic signal that is distant from the meteoric water line (Figure 9).
From a permafrost perspective, the thawing of fossilized water in permafrost-affected rock in large volumes could be related
to a deepening of the active layer and degradation of high mountain permafrost - a regional phenomenon seen in recent decades



in boreholes in the Alps and other mountain ranges (Magnin et al., 2024; Noetzli et al., 2024), but never observed directly in
water samples from fractures.

### 5.4. Implications for alpine geomorphology, hydrogeology, and permafrost

The quantity, timing, and characteristics of water that infiltrates in the fractured, permafrost-affected rocks are important
factors in many geomorphological, hydrological, and geomechanical processes, but our understanding of the parameters
controlling these factors is poor, and the ability to measure them is very limited. For example, the timing and quantity of water
availability from snowmelt are often estimated indirectly using numerical models of energy and water mass balance (Ben-
Asher et al., 2023; Leinauer et al., 2021) and snowpack physics (Lehning et al., 1999; Vionnet et al., 2012). However, the
results of these simulations are highly sensitive to meteorological forcings and hydrogeological parameters that are usually
highly uncertain. The results of this study can profoundly improve the understanding of water availability for infiltration and
the influence of environmental parameters.
The new information we provide on water flow in permafrost rock fractures can also be used to improve coupled heat and
water flow models by providing the parameters needed to calculate heat advection from the surface (flow rate and water
temperature). The average water temperature measured during flow events is 6.1 °C. Theoretical hydrodynamic-thermal
models suggested that the flow of water with temperatures above 5 °C is efficient in thawing permafrost-affected rocks along
fractures (Magnin and Josnin, 2021).
Several recent studies suggested that water-related processes are driving rockwall instability in mountain permafrost (Cathala
et al., 2024; Gruber et al., 2004; Krautblatter et al., 2012; Magnin and Josnin, 2021). Analysis of 1152 rockfall events in the
Mont Blanc massif between 2015-2021 (Magnin et al., 2023; Ravanel and Deline, 2013) show that 96% of the events occurred
between June and September, with the highest numbers in July, before the maximum depth of the seasonal active layer was
reached (Magnin et al., 2023), possibly due to enhanced water flows when snow melts in the early summer.
The cable car to Aiguille du Midi has been operating since 1950's. The staff of the operating company of the site reported that
significant water flow from the fractures in the gallery started in the particularly hot summer of 2015. The reason for the
initiation of the observed seasonal flow is unknown, but it is reasonable to suggest that it is related to the gradual heating of
the rock mass in AdM and the development of the active layer that is observed in monitored boreholes in the site (Magnin et
al., 2024). It points to the possibility that the environmental conditions in AdM are in a transient state, and have reached a
threshold that triggers substantial water availability to fractures in the permafrost rocks.

### 6. Conclusions

This study presents novel, direct observations of water infiltration in a high mountain permafrost rock wall, providing rare
field data on processes that are typically poorly understood and rarely monitored. Using a two-year monitoring system installed




inside man-made tunnels at the Aiguille du Midi (3842 m a.s.l.) in the Mont Blanc massif, we tracked real-time water flow,
temperature, electrical conductivity, and the infiltration of fluorescent tracers injected into the overlying snowpack. These
measurements were combined with GST and meteorological data to investigate the origin, timing, and dynamics of water flow
in permafrost-affected fractured rock.
Our main findings are:
• Water flow in fractures is seasonal and began when AT exceeded 0 °C. Steady flow with daily oscillations started when
GST rose above 0°C, several weeks (in 2022), or several days (in 2023) after the initiation of flow and contained the dye
tracer inserted in the snowpack above the gallery.
• Fluorescent tracers were detected in the first water flow in early summer and in the flow that followed until mid-summer.
This confirms that snowmelt from the winter snowpack is the main source of water.
• In snow-free conditions, during late summer, rain also contributes to the flow.
• The peak flow shows short lag times relative to the peaks of air and ground temperatures (3–9 h and 0–3 h, respectively),
indicating rapid, unsaturated infiltration pathways.
• Evidence from electrical conductivity measurements, stable water isotopes, and analysis of recession curves suggests that
water stored in the rock is contributing to the flow, possibly from the thawing of older ice in the fracture system. The
question of the origin of water with high electrical conductivity and a unique isotopic signature needs further investigation.
However, if confirmed, it would provide direct evidence of the melting of fossil water released with permafrost
degradation.
• Distinct flow regimes of flow collected from two nearby fractures in Box 1 and Box 2 demonstrate a heterogeneous
fracture network with varying sediment infill. It reveals the existence of fractures directly linked to the surface on the one
hand, and fractures with sand infill and likely ice fill with a longer transfer time on the other hand. Moreover, the hydraulic
characteristics of the fractures show unsaturated flow with preferential paths at the end of each warm season before the
active layer freezes again.
• Water temperatures often exceed 5 °C; altogether with the intense water flow measured, it strongly suggest that advective
heat transfer likely contributes to ice melting within the rock mass.
This work offers direct empirical evidence of how surface water infiltrates permafrost rock walls and interacts with the surface
and internal fracture systems. These findings are critical for the development of coupled hydro-thermal models and
understanding how climate warming affects permafrost degradation, water pathways, and slope stability. The approach and
results can help future studies aiming to characterize hydrogeological processes in high-altitude rock fractures, identify early
signs of geomorphic instability, and assess the vulnerability of alpine permafrost landscapes under continued climate change.
**Author contribution**
MB, JYJ, FM: Conceptualization, Data curation, Investigation, Formal analysis, Methodology, Writing.



AC: Data curation, Investigation, Formal analysis, Methodology, Writing.

JB: Investigation, Writing.

EM: Investigation, Methodology.

AP: Resources.

YP: Data curation, Methodology.

**Competing interests**

The authors declare that they have no conflict of interest

**Acknowledgements**

We thank the Compagnie du Mont Blanc and the Aiguille du Midi station staff for their support and access to the site. We also thank UMR 1114 EMMAH laboratory (Environnement Méditerranéen et Modélisation des Agro-Hydrosystèmes) for the stable isotope analysis. We are grateful for the assistance of Marine Quiers in planning and applying fluorescence techniques. This research has been supported by the Agence Nationale de la Recherche (WISPER project, no. ANR-19-CE01-0018).

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
