# Peer review of "Water flow timing, quantity, and sources in a fractured high mountain"

_EGUsphere, 2025_

## Referee Comment (RC1)

**Paper 'Water flow timing, quantity and sources in a fractured high mountain permafrost rock wall' submitted by Ben-Asher et al. (EGUsphere).**

Reviewer: Marcia Phillips

**Detailed comments:**

l. 18: air temperatures (ATs)

l. 18: rock temperatures (not *ground*)

l. 22: flow rates

Key words: water infiltration

Introduction: perhaps you could mention somewhere that water infiltration due to loss of ice plugs is a problem for tourist infrastructures like the AdM, Jungfraujoch or Klein Matterhorn and that the owners have had to install protective roofing in the past decades in the tunnels so the tourists don't get wet (as you use this roofing for your experiment).

l. 34: for an example of water driving a catastrophic failure (Pizzo Cengalo) see Walter et al. 2020 https://doi.org/10.1016/j.geomorph.2019.106933 The references you use here are more process related and not necessarily linked to rock slope failure events.

l. 37: ... and leads to rock fall...

l. 38: ... may also trigger large rock slope failures by reducing...

l. 39: ... the presence of sealing ice in pores and fractures favors...

l. 42: for another example of thermal perturbations (warming and cooling), see https://doi.org/10.1016/j.coldregions.2016.02.010 (Phillips et al. 2016)

l. 51: terrain

l. 51 a reference for hydrological studies in rock glaciers (Bast et al. 2024) https://doi.org/10.5194/tc-18-3141-2024 and in scree slopes (Pellet & Hauck 2017) https://doi.org/10.5194/hess-21-3199-2017

l. 62 infiltration of water

l. 72 elevation (not altitude, which is used when flying)

l. 74 showed

l. 79: fracture network

l. 84: describe the seasonal evolution

l. 85 ... the extent of ice filling or plugging and develop...

l. 96: ... the uplift of which... (not whose)

l. 113 and throughout the paper (and in the figures): replace galleries with tunnels

Figure 1: label the different panels of the figure (a,b,c) and refer to the labels in the caption. Maps provided by the Swiss Federal Office of Topography swisstopo.

l. 121: second warmest years on record.

l. 122 (MAAT)

l. 146 Methods

Figure 3: the top right panel is illegible. What does Location Inf. Elevation refer to and is it needed? What is the pink structure on top (antenna? Building?). Add more information in the figure caption.

l. 157: ... to trace the water source and rate of infiltration... (?)

l. 164: four 4L bottles and six 1.5L bottles .... to prepare the tracer solutions...

l. 165 ... were inserted...

l. 168-169 label the upper and lower terraces in Figure 1

l. 174 ... to protect them (or insulate them) from direct solar radiation...

Table 1: You are not describing the sensor characteristics but their locations (adapt caption)

l. 178: I suggest you use the method described by Staub & Delaloye 2016 https://doi.org/10.1002/ppp.1890 rather than Hansen & Hoelzle 2004.

l. 179 snow has melted / is absent (it is not actively removed).

l. 194 submerged/suspended (not plunged)

l. 203 five measurement values were...

l. 206: ... where sediments sometimes accumulated. (Interesting - did you measure the grain sizes of the sediments?)

l. 209 thunderstorm? Was the problem caused by lightning?

Figure 4: label the different panels and remove 'and issues' from Box 2. Complete the caption and refer to the yellow frame too. The purple frame looks pink.

Figure 5: Please label the panels. The shading for water volume is not legible. Consider placing the photographs in a separate figure

l. 280: this is an example of a reference not appearing (Error! Reference source not found)

Figure 6: I can't distinguish between 2022 and 2023 in the flow volume part.

Figure 7: this figure is very important and interesting and quite illegible (much too small)! Label panels, add description to caption.

l. 304: melt of the winter snowpack

l. 318 and 321: you say daily oscillations but refer to hourly values

l. 319: from 20 July to 10 August... (not the). Please use one form of date consistently. Sometimes you use 3rd and 19th (e.g. l. 332).

l. 344 0.8 here and 0.75 in Figure 8. Which is correct?

l. 379-383 this should be moved to the discussion.

l. 390 ... suggesting that much of the winter and spring snow was gone by...

Section 4.4 Some of this should be moved to the discussion.

l. 430: is this brick wall shown anywhere in a figure? Where/what is the Hellbronner terrace?

l. 435: Values measured (not measurements taken)

Figure 10: the figure caption does not mention probability (y axis). Could you show the tunnel air temperature too?

l. 445: strong weather signals (not climate!) … at both seasonal and…

l.452-453: what about the role of long wave radiation (in the presence of cloud cover)?

l. 472: did you measure the air temperature in the tunnel? Is there an influence from the infrastructure, from the body heat of the tourists or air fluxes from outside/heated buildings?

l. 476: weather (not climate)

l. 483: 'reference source not found', ditto on l. 506, 513

l. 490: could this also be due to the fact that there was very little snow in winter 2021-2022?

l. 499: remove 'from a geomorphological perspective'. It is rather from a geotechnical or cryospheric perspective.

l. 503 remove well-identifying, just use identifying

l. 520: (approx. 3 m apart)

Section 5.3: perhaps you would like to consider the characteristics of the snowpack and the fact that a layer of ice often forms in spring between the snow and the frozen bedrock (see Phillips et al. 2017 https://doi.org/10.1016/j.coldregions.2017.05.010), which may affect the timing of water infiltration into the fracture system.

l. 527: remove direct

l. 546: the melting of fossilized ice (not the thawing of fossilized water). Have you considered dating the water?

l. 569: 1950s

l. 592: melting of older ice (ice melts, ground thaws)

l. 594: melting of fossil ice (not water)

l. 601: suggests

I suggest you add an Outlook section with further possible investigations and open questions.

Figures in general: please use the same font in all figures, improve their legibility, label the panels, describe the figure in the caption.

Please have the English checked before you resubmit.

---

## Referee Comment (RC2)

*Detailed comments to the paper*
**"Water flow timing, quantity, and sources in a fractured high mountain permafrost rock wall"**
*by Matan Ben-Asher, Antoine Chabas, Jean-Yves Josnin, Josué Bock, Emmanuel Malet, Amaël Poulain, Yves Perrette, and Florence Magnin*

*Major/moderate comments.*

1. *Data analysis relies upon a "moving window cross-correlation" scheme. While this is cited at lines 256-258, no explanations on the algorithm are provided. How is the algorithm parametrized in terms of moving window size? How does the choice of the moving window impact on the analysis? The Authors should also carefully describe between which variables are cross-correlations evaluated as this is somehow not clear throughout the manuscript. Finally, the results of the cross-correlation analysis are depicted in a figure included in the supplementary (Figure S3), thus limiting their visibility. I suggest the Author carefully explaining what they did, adding more details about the advantages and the limitation of the method employed, and including these results (onto which the data presentation and discussion then build upon) in the main body of the paper.*

2. *The presentation of the data in the Results section (Section 4) is somehow long, and some parts could be better rendered and communicated to the reader through graphical representations. Data description appears somehow "scattered" as it is divided in many subsections. I suggest increasing the quality of graphical representations (see also comment #3) and shortening data description merging sub-subsections (for example: merge 4.1.x in a single subsection 4.1).*

3. *Increase the overall quality of all figures and associated figure captions. While the dataset collected by the Authors is relevant, the graphical representation of the results is extremely poor. I strongly suggest revising all figures, with particular focus on Figure 7. Here, some y-axis labels are missing. Figure captions are also extremely synthetic, unclear, and/or incomplete. Each caption should fully explain figure content and describe each sub-panel.*

*Minor comments.*

4. *Please carefully revise the use of English language.*
5. *The date format is not consistent throughout the text.*
6. *Some internal references to figures/tables are missing throughout the text (e.g., lines 272, 281 etc).*
7. *Line 15. Replace "fluorescent tracers" with "fluorescence of tracers".*
8. *Line 18. Acronym "AT" has not been defined yet. Avoid acronyms in the abstract for clarity.*
9. *Line 159. What does "original mineral water" mean? Is it water collected from the site?*
10. *Line 162. Replace "new concentrations" with "new solutions". Then, specify concentrations at which solutions are prepared.*
11. *Lines 165-166. A verb is missing in this sentence.*
12. *Line 174. Replace "isolate from" with "isolate them from".*
13. *Line 199. There is a typo, "Acid-Amino-G" should be "Amino-Acid-G".*
14. *Line 234. There is an extra numbering "3.6".*
15. *Lines 241 to 247. Punctuation in the equations is missing.*
16. *Line 263. The sentence "these include data from 109" is somehow incomplete and unclear.*
17. *Line 368. "zero-curtain period" should not be italic.*
18. *Line 421. What do you mean by "corrected at 25°C"?*
19. *Line 441 (caption of Figure 10). "temperature" instead of "T".*

---

## Referee Comment (RC4)

Detailed comments to

**"Water flow timing, quantity, and sources in a fractured high mountain permafrost rock wall"** by Matan Ben-Asher, Antoine Chabas, Jean-Yves Josnin, Josué Bock, Emmanuel Malet, Amaël Poulain, Yves Perrette, and Florence Magni

L 18: AT --> air temperature

L 39: double citation of the same paper. Please correct.

L 48: Scandroglio at al 2020 doesn't seems to be the right citation here. Remove.

L 69: ... data and a snowmelt model to infer *timing and quantities of water flow* and constrain the hydrological pressure in the fractures.

L 69: You should mention here that for the first time they applied recession curves analysis to high alpine bedrock fractures, not later.

L 84: Weber et al 2017 is working on the Matterhorn – what about the Mont Blanc? Please explain.

L 88: Please clearly and precisely define here the research gaps you want to tackle. You might use bullet points or questions. (Remember that timing and quantity of water flow has been already intensively analyzed in Scandroglio et al 2025).

L 96 to 103: Is this description of the MBM relevant in this context? Please reduce and leave only information that is relevant in this context. No need for the whole geotectonical history of MBM. You are just a few meters under the surface, the reader wants to know what is the situation there, and not get confused.

L 118: Add country borders in the small map.

L 121: I suggest the use of local data instead of average data of the whole Europe, that are not necessary very representative for a unique climatic area, like the MBM.

L 122: What you here call AT is MAAT by convention. Please review overall and clarify if you are talking of MAAT or AT (daily or hourly values).

L 125-138: The reader gets completely lost in this paragraph: a lot of numbers and comparisons but it's hard to follow. You don't have to explain every figure. Please rephrase with only important information.

If you improve Figure 2 by adding the long-term comparison (1993-2022) for AT and precipitation, the comparison would be much easier for the reader.

L 131: remove: *"(continuous hourly records started in 1993 in Chamonix)"* – not relevant.

L 135: You talk about spring 2022 in the previous pargraph with other numbers. Why the repetiton? Clarify

L 143: Improve Figure 2: clearly separate data from Chamonix and AdM, add temperatures form Chamonix and long-term averages for both datasets AT /Precipitation

L 148-149-150: *"(Figure 3, Figure 4). Fluorescent dyes were poured into the snowpack on the rock face above the gallery to trace the water. Ground Surface Temperature (GST) sensors were installed on the rock surface, below and around the snowpack where fluorescent dyes were inserted."* These parts are repeated afterwards. Remove here.

L 154: -Figure 3: text overall too small, increase. *"North camera"* not readable because of the color.

L 165 Missing a verb.

L 173: *"in the vicinity"* define more precisely.

L 179: remove *"marks"*

L 191: Provide precise information in the fractures: length, opening, orientation.

L 203: *"five measurements"* of what? since it's a new paragraph, it is not clear what you are talking about. Clarify or unite to previous paragraph.

Figure 4: The yellow box is unclear to me: clarify what you want to show. What is the last picture on the right in the green box showing? please improve caption!

L 252: *"where flow paths could extend for several hundred meters."* You make an assumption that is not included in the cited paper and not based on any scientific data you present. Why do you suppose that?

How did you compute the path length here, at the AdM? I miss clear figures on: the positioning (depth) of your gallery with respect to the surface, the orientation if the clefts in the area, and a clear explanation on how you are computing path length.

From the Photo in Figure 3 it seems there are only a few meters from "Dye1" location and the galley. Add measures to Figure 3 / Text or include a 2D profile of the study area on scale.

L 253: *"widely open, sub vertical"* Quantify. If possible, provide also images. It seems adequate to present here a scan line/ fracture mapping of the tunnel / of the outside.

L 254: *"Additionally, Scandroglio et al. (2025) used a single best-fit curve for their entire dataset, which included only 23 events over eight years. In comparison, we used 93 events for recession curve analysis (out of 144 events that were recorded, see 4.1.5 for more details) over two consecutive.»* This is part of the discussion, move this sentence there.

How do you define events? This can variate from author to author. Add your definition, it seems to me you define an event for each daily cycles, both for rain and snowmelt. On the contrary Scandroglio et al 2025 considered only rain events and clearly defined events: *"An output flow event starts with a sudden increase in the discharge, independent of the starting value, and ends when the flow returns to a value smaller than a threshold … By convention, multiple flow events are classified as one if precipitation interruptions are shorter than 24 h and the resulting hydrograph at the gauges does not reach baseflow status between the two rain event."* Please be precise and correct or remove this comment.

L 256: Please provide a list of all your events including date, duration and start/peak/end discharge.

L 263: 109 what?

L 268: You don't need to refer multiple time to Fig 5 in the same paragraph.

L 272: *"The timing and magnitude of flow differed between Box 1 and Box 2."* From my understanding box 1 and box2 were collecting water from the same cleft - now I'm confused. Are they not installed on the same metal plate on the ceiling? Please clarify the text before and improve the figures to make it clear!

L 272: Reference missing: please next time you submit a version of your article be sure all references are working. This problem is appearing many times in this article and makes work harder for reviewer.

L 276: This image is hard to read. What is "Max" adding as information here? why not removing it? would strongly improve clarity.
I also recommend using bars instead of a line for the volume of water, since it's "per day".
Is flow rate from 1, 2 or both? Clarify.

L 288: (2581 L): Interesting why so much water so late in summer. Is this summer snow melting?

L 289 Figure 6: You squeeze your most valuable data in these small graphs where it is not possible to properly differenciate the lines and read values. Use the whole page width for B and C, A can go alone or be moved to supplementary
- Showing only the period from mid-May to end of September.
- Maybe even rotate and put full page size.

L 291 – Figure 7: Same problem: This is your central Figure but it's very hard to read.

- consider increasing size and turning it 90°
- hourly flow rates are impossible to read at this scale - consider using daily values and make a zoom for hourly - (see Scandroglio et al 2025 fig. 2)
- add here information on the presence of snow, form the camera
- highlight important events with numbers/letters
- differentiate liquid and solid precipitation (using temperature)
- where is the caption????
- Flow rate = 1+2?
Quite some improvements can be achieved here…

L 293: It's impossible to evaluate the numbers you are suggesting from Fig 7. - provide a zoom (example period) and a statistical analysis.

L 294 - Figure S1: missing the y-axis
I find the analysis presented in S1 not scientifically based and the approach questionable. What is your thesis here, what do you want to prove? You are putting together periods with different length and making an average "curve". How did you decide the length of each period?
It would be much better to provide a statistical analysis. Alternatively, you could plot all cycles together and produce a density graph.

L 294: *"two orders of magnitude lower during the morning time"* where do I see that? please show

L 295: This table is uncomplete. Provide a proof for these values and further statistical information, like standard deviation? Is there a change with time (May to July to September) ?

L 298: S3 should be S2. How did you compute the values (3-9 and 0-3)? Demonstrate how you obtained these values. In the graph for GST the green shape covers the range 0-6.

L 305: *"Most likely fell as rain since the AT was positive«*. Temperature at the end of July 2022is under zero for quite some time (2-3 days... it's impossible to decipher form the picture). Please rephrase correctly.

L 328: *"This event was unusual because it lasted for 3 days"* It is worth to provide a detailed Figure on this event and other special events you talk about, at least in the supporting material. Please add.

L 336 same here

L 338: I suggest moving this information to a table, together with the numbers from 2022, for clarity and comparison.

L 340: 1033 L/day --> similar to Scandroglio et al 2025, comment in the discussion.

L 343: Refer here to fig S2. You extracted the flow recession and then fitted it. Why is just applied on some days and not on others? How do you select which day to fit and which not?  Is this based on how well your recession is fitting? ("...values below 0.8..."). Sorry but if you reject the low values, it's not a surprise that you get high $R^2$ values.

Why are some flow recessions not starting from the maximum value? By selecting arbitrarily, the starting point of your flow recession you strongly influence the results.

From chapter 3.6.1 I understand that you rejected about 1/3 of your events. This is an elevated number, and they cannot be considered just as "outliers". I suggest reviewing the methodology used in this analysis, since these results are very important in your discussion and conclusions.

L 344: Interesting analysis. I see the trends you describe the ranges you provide are not representing what it is shown in the picture.

a = 7-10 to 8.5-5

b = 1-1.08 to 1.04-1.15

L 349: 0.75 ... I read in the text 0.8. Use the same number.

L 362: The link with snow melting is clear, not only a hind. Rephrase.

L 364: *"Summer precipitation episodes are suggested when ..."* Why don't you detect precipitation periods by using AT? it seems more obvious and direct for me

L 385 Repetition, check and remove.

L 403-404: Move to discussion

L 410 How do you explain the "*extended residence time*"?

L 415-416: move to discussion

L 438 – Figure 10: I very appreciate this analysis. Could you also provide the changes of water temperature in time? Why is there a peak at -8°C for AT?

L 455: "*direct precipitation*": what is direct precipitation? When is it indirect?

L 458: *"we conclude that water flow processes in high mountain rock faces are therefore seasonal,»* This is quite an obvious conclusion, not really a novelty. Your results are applicable "close to the surface" but not to bigger depth.

L 461: "*Show that snowmelt is the main source of water in the fractures during the early and main stages of flow, and contributes most of the water*." Similar result to Scandroglio et al 2025.

L 463 to 468: These are results, please move this paragraph to a more adequate position.

L 475: Definition of **heat wave**: "a prolonged period of unusually hot weather." Please clearly define which periods are "heat waves" for you, also with figures.

L 484: "Assuming that water…" Why assuming when you have measurements of water temperature! Show them in the supplementary material.

L 488: remove "likely"

L 494: What do you mean with "transfer rate" explain.

L 459 *"flow is unsaturated"* If the flow stops, there is no water moving, neither saturated nor unsaturated. Please rephrase.

L 496: *"This unsaturated flow shows that there are preferential flow paths into the fractures, leading to open paths available for the melting water of the snowpack in the following spring."* This is an assumption, prove it.

L 506: *"One reason for the delayed flow in Box 2 could be linked to the location of the draining area closer to the colder north face, while the draining area of Box 1 is closer to the west face, which is exposed to more solar radiation."* You can easily prove this by reporting the orientation of the fractures where you measure water flow.

L 519: Similar to Scandroglio et al 2025

L 529: Please add some references.

L 537: "his change in recession form, from aquifer-type (Equation 1) to channel-type (Equation 3) can be explained by the thawing of ice in wide sub-vertical fractures that are likely to react more individually (rather than as a network) and enable rapid flow in the fractured granite". This is a great finding, very interesting! To prove this hypothesis, It would be very important to see this change directly … e.g. by comparing some of the recession curves. I suggest adding a figure, here in the discussion.

L 555: Leinauer et al 2021 is actually using the model of Lehning et al 1999 : SNOWPACK

L 556: Meteorological forcing is driving the software SNOWPACK, therefore simulations are sensitive to it. Rephrase this sentence.

It seems to me that that hydrogeological parameters are highly uncertain also here. Rephrase.

L 560: which flow rate are you suggesting as a parameter for models? quantify it

L 567: *"..the highest numbers in July"*... any connection with your flow peaks 400 L/d and 1000 L/d ???

L 590: Why is this time indication unsaturated path? Where are you discussing this in the text?

L 615: What about data availability?

---

## Author Comment (AC1)

25 September 2025

*We sincerely thank Marcia Phillips for the thorough and constructive review. The comments have helped us to improve the clarity, presentation, and scientific depth of our manuscript. Below, we provide a point-by-point response.* Reviewer comments are reproduced in *black, followed by our responses in italic blue text. All line numbers refer to the original version*. *All the revised figures and captions are available at the bottom of this document.*

**Paper 'Water flow timing, quantity and sources in a fractured high mountain permafrost rock wall' submitted by Ben-Asher et al. (EGUsphere).**

Reviewer: Marcia Phillips

**General comments:**

The paper entitled 'Water flow timing, quantity, and sources in a fractured high mountain permafrost rock wall' by Ben-Asher et al. presents the results of a two-year campaign monitoring fracture water in high elevation permafrost at the Aiguille du Midi, France. The subject is currently of great interest, as rock slope failures in high mountain areas appear to be linked to the loss of sealing permafrost ice plugs in rock fractures and to deep-seated infiltration of water into the newly accessible fracture systems. The extent of the fracture systems and their hydrology is poorly known. This study uses a combination of methods to identify the sources of water flowing through rock fractures, the rates and timing of flow, preferential flow paths, and the thermal regimes of the rock and water. Most of the relevant literature is cited (see my suggestions in the detailed comments (attached) for further literature), but in some cases the references do not appear (error message). Most of the figures need enlarging and labelling to improve their legibility. The figure captions do not adequately describe the figures. The language is mostly clear but with some grammatical or consistency issues (see detailed comments). Some small changes to the paper structure should be considered, particularly in sections 4.3.1 and 4.4, where the explanations/hypotheses should be moved to the Discussion. The paper is highly relevant and I suggest it be accepted for publication, with major modifications.

*The missing literature was added to the manuscript. Erroneous references were corrected.*

*All figures, excluding two, were edited and are now clearer with more informative and detailed captions:*
*Figure 1: Panels were labeled, and the caption refers to each panel specifically.*
*Figure 3: Unnecessary details were omitted. Text size was enlarged. Caption was edited.*
*Figure 4: Panels were labeled, text in the figure was refined. Caption was elaborated with an improved description of the experiment setup.*

*Figure 5: Panels were labeled, and the transparency of the water volume and air temperature colors were reduced to improve visibility.*
*Figure 6: Column bars of the monthly flow volume were widened to improve the distinction between 2022 and 2023 data.*
*Figure 7: Caption was edited and is now much more detailed. Missing y-axis labels were inserted. Panels were labeled.*
*Figure 8: A new figure that was in the supplementary materials. Added with labeled panels and detailed caption.*
*Figure 9: Panels were labeled, and the caption refers to each panel specifically.*
*Figure 10: No changes*
*Figure 11: Caption was edited. Text size increased. A new distribution was added – tunnel air temperature.*

**Detailed comments:**

l. 18: air temperatures (ATs). *Corrected.*

l. 18: rock temperatures (not *ground*). *Corrected.*

l. 22: flow rates. *Corrected.*

Key words: water infiltration. *Keyword added..*

Introduction: perhaps you could mention somewhere that water infiltration due to loss of ice plugs is a problem for tourist infrastructures like the AdM, Jungfraujoch or Klein Matterhorn and that the owners have had to install protective roofing in the past decades in the tunnels so the tourists don't get wet (as you use this roofing for your experiment). *We appreciate the reviewer's suggestion to highlight the implications of water infiltration for tourist infrastructure. However, in the context of this paper, our focus is on the hydrological and permafrost-related processes and their relevance for landscape evolution and slope stability. While tourist comfort (e.g., avoiding water dripping inside tunnels) is a practical concern, it is not scientifically relevant in the framework of Hydrology and Earth System Sciences. For this reason, we chose not to include such details and instead emphasized the broader geomorphological and hydrothermal implications of our observations.*

l. 34: for an example of water driving a catastrophic failure (Pizzo Cengalo) see Walter et al. 2020 https://doi.org/10.1016/j.geomorph.2019.106933 The references you use here are more process related and not necessarily linked to rock slope failure events. *Reference was added.*

l. 37: … and leads to rock fall… . *Added to text.*

l. 38: … may also trigger large rock slope failures by reducing… *Corrected.*

l. 39: … the presence of sealing ice in pores and fractures favors… *Corrected.*

l. 42: for another example of thermal perturbations (warming and cooling), see https://doi.org/10.1016/j.coldregions.2016.02.010 (Phillips et al. 2016). *Reference was added.*

l. 51: terrain. *Corrected.*

l. 51 a reference for hydrological studies in rock glaciers (Bast et al. 2024) https://doi.org/10.5194/tc-18-3141-2024 and in scree slopes (Pellet & Hauck 2017) https://doi.org/10.5194/hess-21-3199-2017. *References were added.*

l. 62 infiltration of water. *Corrected.*

l. 72 elevation (not altitude, which is used when flying). *corrected here and in the conclusion chapter.*

l. 74 showed. *Corrected.*

l. 79: fracture network. *Corrected.*

l. 84: describe the seasonal evolution. *Description added: " ...with reversible opening in winter, superimposed on a long-term irreversible opening trend".*

l. 85 ... the extent of ice filling or plugging and develop... T*ext added.*

l. 96: ... the uplift of which... (not whose). *Corrected.*

l. 113 and throughout the paper (and in the figures): replace galleries with tunnels. *Replaced all.*

Figure 1: label the different panels of the figure (a,b,c) and refer to the labels in the caption. Maps provided by the Swiss Federal Office of Topography swisstopo. *Panels were labeled, and the caption refers to each panel specifically.*

l. 121: second warmest years on record. *Corrected.*

l. 122 (MAAT). *No need to use MAAT initials since it is only mentioned once in the text.*

l. 146 Methods. *Corrected.*

Figure 3: the top right panel is illegible. What does Location Inf. Elevation refer to and is it needed? What is the pink structure on top (antenna? Building?). Add more information in the figure caption. *Unnecessary details were omitted. Text size was enlarged. Caption was edited.*

l. 157: ... to trace the water source and rate of infiltration... (?). *Added text.*

l. 164: four 4L bottles and six 1.5L bottles .... to prepare the tracer solutions... *Added.*

l. 165 ... were inserted... *Updated the text.*

l. 168-169 label the upper and lower terraces in Figure 1. *Figure 1 does not show the terraces. Labels were added in Figure 3 instead.*

l. 174 ... to protect them (or insulate them) from direct solar radiation... *Sentence updated.*

Table 1: You are not describing the sensor characteristics but their locations (adapt caption). *Updated the caption.*

l. 178: I suggest you use the method described by Staub & Delaloye 2016 https://doi.org/10.1002/ppp.1890 rather than Hansen & Hoelzle 2004. *The citation of Hansen & Hoelzle 2004 is meant to support the assumption that stagnant GST ~0°C represents a melting period. The same assumption is made by Staub & Delaloye 2016. Reference added.*

l. 179 snow has melted / is absent (it is not actively removed). *Text updated.*

l. 194 submerged/suspended (not plunged). *Replaced word.*

l. 203 five measurement values were... *Corrected.*

l. 206: ... where sediments sometimes accumulated. (Interesting - did you measure the grain sizes of the sediments?). *Sentence updated. Unfortunately, the sediments were not collected during the experiment period.*

l. 209 thunderstorm? Was the problem caused by lightning? *Cable car was not operating because of the storm, for safety reasons.*

Figure 4: label the different panels and remove 'and issues' from Box 2. Complete the caption and refer to the yellow frame too. The purple frame looks pink. *We labeled the panels, corrected the text boxes, and edited the caption.*

Figure 5: Please label the panels. The shading for water volume is not legible. Consider placing the photographs in a separate figure. *Labeled the panels, edited the caption. did not put the photos in a separate figure but separated them into defined panels.*

l. 280: this is an example of a reference not appearing (Error! Reference source not found). *This error was fixed here and the rest of the text.*

Figure 6: I can't distinguish between 2022 and 2023 in the flow volume part. *Bars were made wider to improve the difference between the two filling textures of 2022 and 2023.*

Figure 7: this figure is very important and interesting and quite illegible (much too small)! Label panels, add description to caption. *We increased the figure size, added panel labels, and the description in the caption text.*

l. 304: melt of the winter snowpack and. *Replaced word.*

l. 318 and 321: you say daily oscillations but refer to hourly values. *In oscillations, we refer to the period that a waveform completes a cycle, i.e. peak-to-peak.*

l. 319: from 20 July to 10 August... (not the). Please use one form of date consistently. Sometimes you use 3rd and 19th (e.g. l. 332). *Sentence corrected. All dates in the text were edited to a unified format.*

l. 344 0.8 here and 0.75 in Figure 8. Which is correct? *0.8 is the correct value. Updated in the text.*

l. 379-383 this should be moved to the discussion. *Moved to section 5.3 in the discussion chapter.*

l. 390 ... suggesting that much of the winter and spring snow was gone by... *Updated the text.*

Section 4.4 Some of this should be moved to the discussion. *We agree. Moved the section 5.3.*

l. 430: is this brick wall shown anywhere in a figure? Where/what is the Hellbronner terrace? *We changed the description to: "at another location in a tunnel under the north-east face of the central peak, near the exit of the cable car going to Pointe Helbronner (Italy)."*

l. 435: Values measured (not measurements taken). *Corrected.*

Figure 10: the figure caption does not mention probability (y axis). Could you show the tunnel air temperature too? *Updated the caption and added the tunnel air temperature distribution.*

l. 445: strong weather signals (not climate!) ... at both seasonal and... *Corrected*

l.452-453: what about the role of long wave radiation (in the presence of cloud cover*)? This sentence describes an observation in the measurements. Long wavelength radiation could have increased the air temperature and increase the surface heat flux directly, but there is no data to support it.*

l. 472: did you measure the air temperature in the tunnel? Is there an influence from the infrastructure, from the body heat of the tourists or air fluxes from outside/heated buildings? *We did measure air T in the tunnel. The temperature distribution was added to Figure 11. The following sentence was added: In addition, the touristic infrastructure and human presence can contribute internal heat sources, including heating systems, the elevator motor, and body heat from visitors.*

l. 476: weather (not climate). *Corrected.*

l. 483: 'reference source not found', ditto on l. 506, 513. *This error was solved.*

l. 490: could this also be due to the fact that there was very little snow in winter 2021-2022? *Yes, absolutely. It is actually seen in Figure 5 when comparing snow cover in mid-June in both years. Added to the text in section 5.1.1.*

l. 499: remove 'from a geomorphological perspective'. It is rather from a geotechnical or cryospheric perspective. *Deleted the sentence.*

l. 503 remove well-identifying, just use identifying. *Removed.*

l. 520: (approx. 3 m apart). *Corrected.*

Section 5.3: perhaps you would like to consider the characteristics of the snowpack and the fact that a layer of ice often forms in spring between the snow and the frozen bedrock (see Phillips et al. 2017 https://doi.org/10.1016/j.coldregions.2017.05.010), which may affect the timing of water infiltration into the fracture system. *We are familiar with the work of Phillips et al. (2017), and we acknowledge that ice layers at the snow–ground interface can influence infiltration in some settings. However, our results show no evidence of such an effect in this study. The first flow events occurred in direct correlation with surface warming above the*

*melting point and already contained tracer dye applied at the snow surface, indicating rapid infiltration and high connectivity between the snowpack and the fracture system.*

l. 527: remove direct. *Removed.*

l. 546: the melting of fossilized ice (not the thawing of fossilized water). Have you considered dating the water? *Thank you for this comment. We agree that the melting of fossil ice is a more accurate term. We used stable isotopes to attempt to differentiate modern from older water, but the results were not conclusive. Absolute dating of meteoric water is not straightforward, yet it is certainly worth considering in future work, especially in light of our findings..*

l. 569: 1950s. *Updated the text.*

l. 592: melting of older ice (ice melts, ground thaws). *Updated the text.*

l. 594: melting of fossil ice (not water). *Updated the text.*

l. 601: suggests. *Corrected.*

I suggest you add an Outlook section with further possible investigations and open questions.*This is an excellent idea. We added a new section 5.5. Outlook and Future Directions: Future investigations could build upon this study by applying more detailed chemical analyses of dissolved elements, which would help constrain water–rock interaction processes and potential solute sources. Characterizing the mineralogy and size distribution of sediments flushed from fractures could provide complementary evidence on transport pathways and mechanical erosion. Further stable isotope analyses, combined with absolute dating techniques (e.g., tritium–helium, radiocarbon, or noble gas methods), may allow a clearer distinction between modern meltwater, rain inputs, and contributions from older subsurface ice. Together, these approaches would refine our understanding of fracture-scale hydrology in steep permafrost rock walls and its sensitivity to climate forcing.*

Figures in general: please use the same font in all figures, improve their legibility, label the panels, describe the figure in the caption. *The figures were edited and are now clearer and informative.*

Please have the English checked before you resubmit. *Done*

*We believe that the revised manuscript fully addresses the reviewer's comments. In particular, we have restructured some sections, improved all figures and captions, standardized terminology, corrected errors in references, and revised the English throughout. We also added a new section on Outlook and Future Directions to highlight open questions and potential avenues for future work. We thank Marcia Phillips again for the valuable feedback, which has significantly strengthened our manuscript.*

25 September 2025

*Figures*

[Figure]

**Figure 1: A**) Location of the Aiguille du Midi in the Mont Blanc massif. B) view of the three peaks at Aiguille du Midi. (Picture: S. Gruber). C) Location of the Mont Blanc massif on the border of France, Italy and Switzerland. Maps provided by the Swiss Federal Office of Topography swisstopo.

[Figure]

**Figure 2: Daily (thin lines) and monthly (thick lines) air temperature and weekly precipitation in 2022 and 2023 (bars). Air temperature was measured at the top of the Aiguille du Midi and precipitation was measured in Chamonix (1042 m asl). Data provided by Météo France.**

[Figure]

**Figure 3: Sketch of the methodological approach to track and monitor water flows in the Aiguille du Midi central pillar. Note the location of the insertion locations of the dye tracers in the snowpacks on the terraces above the water monitoring boxes.**

[Figure]

**Figure 4: Real-time monitoring system in the tunnel. A) Metal roof draining to Box 1 (pink dashed frame). B) A 3D printed siphon that was placed directly under the water output from the fracture, quipped with T and conductivity sensors (yellow dashed frame). C) Box 1 interior with rain gauge to monitor flow rate, and a sampling bottle and bucket. D) Box 2 with sediments (green dashed frame). E) Fluorescence probe by TRAQUA located in the a specially designed siphon for continuous real time monitoring of the dye tracers.**

[Figure]

**Figure 5: A)** Photos showing the evolution of the snow cover on the NE face during the thawing seasons in 2022. Note the change in snow cover. **B)** 2022 season AT, GST measured on the NE face, above the tunnel entrance, directly above the monitoring system, and flow rate measured at output from rock fractures in the tunnel wall. Solid lines represent the daily mean. **C)** 2023 season AT, GST measured on the NE face, above the tunnel entrance, directly above the monitoring system, and flow rate measured at output from rock fractures in the tunnel wall. Solid lines represent the daily mean. Note the zero curtain period which marks the thawing of the snowpack and exposure of the rock surface to atmospheric heating. **D) )** Photos showing the evolution of the snow cover on the NE face during the thawing seasons in 2023.

[Figure]

**Figure 6: A) Monthly distribution of flow volume in Box 1 and Box 2 during the 2022 and 2023 seasons. B) Measured flow rate vs. time in 2022. C) Measured flow rate vs. time in 2023.**

[Figure]

**Figure 7: Annual time series. A) Air temperature (AT) measured by Météo-France in Aiguille du Midi. B) Ground surface temperatures (GST) measured using iButtons at the rock surface on rock slope above the water collecting system, near the location of fluorescent dyes injection. C-D) Flow rate measured in both box 1 + box 2 (purple) and daily precipitation measured in Chamonix meteorolocical station (Météo-France) (yellow bars). E) Normalized fluorescence signal of amino acid-G dye tracer that was inserted in the upper terrace in both seasons (2022 and 2023). F) Normalized fluorescence signal of Sulphorhodamins-B (inserted in 2022) and Fluorescence (inserted in 2023) dye tracers. The Sulphorhodamins-B dye was never detected. G) Water conductivity that was monitored continuously at the outlet of water from the fracture in the tunnel.**

[Figure]

**Figure 8: Results of moving-window cross-correlation analysis of water flow with (A) air temperatures (AT) and (B) ground surface temperatures (GST), during 2022 season. The horizontal axis represents the days, and the vertical axis represents the size of the lag time, in hours, between the flow rate time series with AT (upper plot) and GST (lower plot). The color bar represents the value of the Pearson correlation coefficient (PCC) (1: high correlation, 0: no correlation, -1: reverse correlation). The green frame marks the range of lag times that show high PCC. Results of the cross-correlation analysis of 2023 season show similar results and can be found in the supplementary materials, in figure S3.**

[Figure]

**Figure 9: Values of the recession curve exponential coefficients a (top) and b (bottom) in 2022 (left) and 2023 (right) vs. time. A) values of the 'a' coefficient of the recession curves of flow events in 2022 in box 1 (purple circles) and box 2 (yellow circles). B) values of the 'a' coefficient of the recession curves of flow events in 2023 in box 1 (purple circles) and box 2 (yellow circles). C) values of the 'b' coefficient of the recession curves of flow events in 2022 in box 1 (purple circles) and box 2 (yellow circles). D) values of the 'b' coefficient of the recession curves of flow events in 2023 in box 1 (purple circles) and box 2 (yellow circles). Values obtained from curves with R² values below 0.8 were omitted from the analysis. The black line is the linear regression of all the points (box 1 + box 2) with ±standard error (dashed black lines). The vertical lines indicate the timing of the detection of the fluorescent dye in the water that exits the fractures (green), the rapid drop of the signal intensity (orange), and the disappearance of the signal (red).**

[Figure]

**Figure 10: Stable isotopes δ¹⁸O and δD in water samples. Note the two outliers (labeled in red) from the global meteoric water line (GMWL, black line) in samples taken from Box 2 on 28 July 2022.**

[Figure]

**Figure 11: Probability distribution of temperatures monitored during flow events (blue), atmospheric ATs (orange), ground surface temperatures (yellow), and tunnel wall (green). All distributions show data from the thawing season in 2022 and 2023 (15 May – 15 September). Note that the water temperature distribution (blue) shows only data when water flow was detected in the monitoring system, while the other temperature distributions represent the entire data within the thawing season.**

---

## Author Comment (AC2)

25 September 2025

We thank Reviewer 2 for the thorough and constructive review. The comments helped us improve the clarity, structure, and presentation of our manuscript. Below we provide a detailed, point-by-point response. Reviewer comments are reproduced in italics, and our responses follow each comment. All line numbers refer to the original version of the manuscript. All the revised figures and captions are available at the bottom of this document.

*Detailed comments to the paper*
*"**Water flow timing, quantity, and sources in a fractured high mountain permafrost rock wall**"*
*by Matan Ben-Asher, Antoine Chabas, Jean-Yves Josnin, Josué Bock, Emmanuel Malet, Amaël*
*Poulain, Yves Perrette, and Florence Magnin*

***Major/moderate comments.***
1. *Data analysis relies upon a "moving window cross-correlation" scheme. While this is cited at lines 256-258, no explanations on the algorithm are provided. How is the algorithm parametrized in terms of moving window size? How does the choice of the moving window impact on the analysis? The Authors should also carefully describe between which variables are cross-correlations evaluated as this is somehow not clear throughout the manuscript. Finally, the results of the cross-correlation analysis are depicted in a figure included in the supplementary (Figure S3), thus limiting their visibility. I suggest the Author carefully explaining what they did, adding more details about the advantages and the limitation of the method employed, and including these results (onto which the data presentation and discussion then build upon) in the main body of the paper.*

Following this important comment, we added a more detailed explanation of the method in section 3.5.2. We also inserted a new figure (now Figure 8) that includes the results of the cross-correlation analysis for the 2022 season. The 2023 season is presented in the supplementary materials - Figure S3.

2. *The presentation of the data in the Results section (Section 4) is somehow long, and some parts could be better rendered and communicated to the reader through graphical representations. Data description appears somehow "scattered" as it is divided in many subsections. I suggest increasing the quality of graphical representations (see also comment #3) and shortening data description merging sub-subsections (for example: merge 4.1.x in a single subsection 4.1).*

The results section was significantly shortened, mostly by reducing the details of flow behaviour. All subsection 4.1.x were merged into the main 4.1 section, as suggested. All figures were edited to improve their resolution, increased text size where needed, labeled panels and more detailed caption text.

3. *Increase the overall quality of all figures and associated figure captions. While the datase tcollected by the Authors is relevant, the graphical representation of the*

*results is extremely poor. I strongly suggest revising all figures, with particular focus on Figure 7. Here, some y-axis labels are missing. Figure captions are also extremely synthetic, unclear, and/or incomplete. Each caption should fully explain figure content and describe each sub-panel.*

All the figures were edited and are now clearer with more informative and detailed captions:
Figure 1: Panels were labeled, and the caption refers to each panel specifically.
Figure 3: Unnecessary details were omitted. Text size was enlarged. Caption was edited.
Figure 4: Panels were labeled, text in the figure was refined. Caption was elaborated with an improved description of the experiment setup.
Figure 5: Panels were labeled, and the transparency of the water volume and air temperature colors were reduced to improve visibility.
Figure 6: Column bars of the monthly flow volume were widened to improve the distinction between 2022 and 2023 data.
Figure 7: Caption was edited and is now much more detailed. Missing y-axis labels were inserted. Panels were labeled.
Figure 8: A new figure that was in the supplementary materials. Added with labeled panels and detailed caption.
Figure 9: Panels were labeled, and the caption refers to each panel specifically.
Figure 10: No changes
Figure 11: Caption was edited. Text size increased. A new distribution was added – tunnel air temperature.

*Minor comments.*
*4. Please carefully revise the use of English language.* The text was revised and edited.
*5. The date format is not consistent throughout the text.* All dates were edited to a single format.
*6. Some internal references to figures/tables are missing throughout the text (e.g., lines 272, 281 etc).* This error was fixed.
*7. Line 15. Replace "fluorescent tracers" with "fluorescence of tracers".* Done.
*8. Line 18. Acronym "AT" has not been defined yet. Avoid acronyms in the abstract for clarity.* Done
*9. Line 159. What does "original mineral water" mean? Is it water collected from the site?* It means that the original mineral water content that came in the bottles was used to produce the tracer solution. The sentence was edited for clarity: The solutions were prepared and carried in "Ondine®"
*10. Line 162. Replace "new concentrations" with "new solutions". Then, specify concentrations at which solutions are prepared.* Done.
*11. Lines 165-166. A verb is missing in this sentence.* Edited the sentence: The dyes powders were inserted directly into the bottles with the original mineral water.
*12. Line 174. Replace "isolate from" with "isolate them from".* Edited the sentence: The holes with the coin-sized sensors were filled with gray polymer clay to insulate the sensors from direct solar radiation on the metal sensor.

*13. Line 199. There is a typo, "Acid-Amino-G" should be "Amino-Acid-G".* Corrected.

*14. Line 234. There is an extra numbering "3.6".* Deleted.

*15. Lines 241 to 247. Punctuation in the equations is missing.* Punctuation was added.

*16. Line 263. The sentence "these include data from 109" is somehow incomplete and unclear.* Deleted.

*17. Line 368. "zero-curtain period" should not be italic.* Done

*18. Line 421. What do you mean by "corrected at 25°C"?* Since water conductivity is influenced by temperature, the values are normally reported after correction to room temperature. The sentence was rephrased: "…after correction to a standard temperature of 25°C."

*19. Line 441 (caption of Figure 10). "temperature" instead of "T".* Done.

We believe that these revisions address all the reviewer's concerns and have substantially improved the manuscript. We thank Reviewer 2 again for their constructive feedback and valuable suggestions.

Figures

[Figure]

**Figure 1: A**) Location of the Aiguille du Midi in the Mont Blanc massif. B) view of the three peaks at Aiguille du Midi. (Picture: S. Gruber). C) Location of the Mont Blanc massif on the border of France, Italy and Switzerland. Maps provided by the Swiss Federal Office of Topography swisstopo.

[Figure]

Figure 2: Daily (thin lines) and monthly (thick lines) air temperature and weekly precipitation in 2022 and 2023 (bars). Air temperature was measured at the top of the Aiguille du Midi and precipitation was measured in Chamonix (1042 m asl). Data provided by Météo France.

[Figure]

**Figure 3: Sketch of the methodological approach to track and monitor water flows in the Aiguille du Midi central pillar. Note the location of the insertion locations of the dye tracers in the snowpacks on the terraces above the water monitoring boxes.**

[Figure]

**Figure 4: Real-time monitoring system in the tunnel. A) Metal roof draining to Box 1 (pink dashed frame). B) A 3D printed siphon that was placed directly under the water output from the fracture, quipped with T and conductivity sensors (yellow dashed frame). C) Box 1 interior with rain gauge to monitor flow rate, and a sampling bottle and bucket. D) Box 2 with sediments (green dashed frame). E) Fluorescence probe by TRAQUA located in the a specially designed siphon for continuous real time monitoring of the dye tracers.**

[Figure]

**Figure 5:** A) Photos showing the evolution of the snow cover on the NE face during the thawing seasons in 2022. Note the change in snow cover. B) 2022 season AT, GST measured on the NE face, above the tunnel entrance, directly above the monitoring system, and flow rate measured at output from rock fractures in the tunnel wall. Solid lines represent the daily mean. C) 2023 season AT, GST measured on the NE face, above the tunnel entrance, directly above the monitoring system, and flow rate measured at output from rock fractures in the tunnel wall. Solid lines represent the daily mean. Note the zero curtain period which marks the thawing of the snowpack and exposure of the rock surface to atmospheric heating. D) ) Photos showing the evolution of the snow cover on the NE face during the thawing seasons in 2023.

[Figure]

**Figure 6: A) Monthly distribution of flow volume in Box 1 and Box 2 during the 2022 and 2023 seasons. B) Measured flow rate vs. time in 2022. C) Measured flow rate vs. time in 2023.**

[Figure]

**Figure 7: Annual time series. A)** Air temperature (AT) measured by Météo-France in Aiguille du Midi. **B)** Ground surface temperatures (GST) measured using iButtons at the rock surface on rock slope above the water collecting system, near the location of fluorescent dyes injection. **C-D)** Flow rate measured in both box 1 + box 2 (purple) and daily precipitation measured in Chamonix meteorolocical station (Météo-France) (yellow bars). **E)** Normalized fluorescence signal of amino acid-G dye tracer that was inserted in the upper terrace in both seasons (2022 and 2023). **F)** Normalized fluorescence signal of Sulphorhodamins-B (inserted in 2022) and Fluorescence (inserted in 2023) dye tracers. The Sulphorhodamins-B dye was never detected. **G)** Water conductivity that was monitored continuously at the outlet of water from the fracture in the tunnel.

[Figure]

**Figure 8: Results of moving-window cross-correlation analysis of water flow with (A) air temperatures (AT) and (B) ground surface temperatures (GST), during 2022 season. The horizontal axis represents the days, and the vertical axis represents the size of the lag time, in hours, between the flow rate time series with AT (upper plot) and GST (lower plot). The color bar represents the value of the Pearson correlation coefficient (PCC) (1: high correlation, 0: no correlation, -1: reverse correlation). The green frame marks the range of lag times that show high PCC. Results of the cross-correlation analysis of 2023 season show similar results and can be found in the supplementary materials, in figure S3.**

[Figure]

**Figure 9: Values of the recession curve exponential coefficients a (top) and b (bottom) in 2022 (left) and 2023 (right) vs. time. A) values of the 'a' coefficient of the recession curves of flow events in 2022 in box 1 (purple circles) and box 2 (yellow circles). B) values of the 'a' coefficient of the recession curves of flow events in 2023 in box 1 (purple circles) and box 2 (yellow circles). C) values of the 'b' coefficient of the recession curves of flow events in 2022 in box 1 (purple circles) and box 2 (yellow circles). D) values of the 'b' coefficient of the recession curves of flow events in 2023 in box 1 (purple circles) and box 2 (yellow circles). Values obtained from curves with $R^2$ values below 0.8 were omitted from the analysis. The black line is the linear regression of all the points (box 1 + box 2) with ±standard error (dashed black lines). The vertical lines indicate the timing of the detection of the fluorescent dye in the water that exits the fractures (green), the rapid drop of the signal intensity (orange), and the disappearance of the signal (red).**

[Figure]

**Figure 10: Stable isotopes δ¹⁸O and δD in water samples. Note the two outliers (labeled in red) from the global meteoric water line (GMWL, black line) in samples taken from Box 2 on 28 July 2022.**

[Figure]

**Figure 11: Probability distribution of temperatures monitored during flow events (blue), atmospheric ATs (orange), ground surface temperatures (yellow), and tunnel wall (green). All distributions show data from the thawing season in 2022 and 2023 (15 May – 15 September). Note that the water temperature distribution (blue) shows only data when water flow was detected in the monitoring system, while the other temperature distributions represent the entire data within the thawing season.**